# An analysis of offshore wind farm SCADA measurements to identify key parameters influencing the magnitude of wake effects

Niko Mittelmeier[1], Julian Allin[1], Tomas Blodau[1], Davide Trabucchi[2], Gerald Steinfeld[2], Andreas Rott[2] and Martin Kühn[2]

[1] Senvion GmbH, Überseering 10, 22297 Hamburg, Germany
[2] ForWind – University of Oldenburg, Institute of Physics, Küpkersweg 70, 26129 Oldenburg

*Correspondence to*: Niko Mittelmeier (niko.mittelmeier@senvion.com)

**Abstract.** For offshore wind farms wake effects are among the largest sources for losses in energy production. At the same time wake modelling is still associated with very high uncertainties. Therefore current research focusses on improving wake model predictions. It is known that atmospheric conditions, especially atmospheric stability, crucially influence the magnitude of those wake effects. The classification of atmospheric stability is usually based on measurements from met masts, buoys or LiDARs (Light Detection and Ranging). In offshore conditions these measurements are expensive and scarce. However, every wind farm permanently produces SCADA (Supervisory Control and Data Acquisition) measurements. The objective of this study is to establish a classification for the magnitude of wake effects based on SCADA data. This delivers a basis to fit engineering wake models better to the ambient conditions in an offshore wind farm. The method is established with data from two offshore wind farms, which have a met mast nearby each. A correlation is established between the stability classification from the met mast and signals within the SCADA data from the wind farm. The significance of these new signals on power production is demonstrated with data from two wind farms with met mast and long range LiDAR measurements. Additionally the method is validated with data from another wind farm without met mast. The proposed signal consists of a good correlation between the standard deviation of active power divided by the average power of wind turbines in free flow, with the ambient turbulence intensity TI, when the wind turbines were operating in partial load. It allows to distinguish between conditions with different magnitude of wake effects. The proposed signal is very sensitive to increased turbulence induced by neighbouring turbines and wind farms, even at a distance of more than 38 rotor diameters.

## 1 Introduction

Wake effects are one of the largest sources of losses in offshore energy yield assessment. This makes wake modelling very important and much research is ongoing to improve wake model predictions. In the latest offshore CREYAP benchmark exercise (Comparative Resource and Energy Yield Assessment Procedure) wake modelling was found to be the prediction with the highest variation among the participants (Mortensen et al., 2015).

In order to be able to use a wake model for validating the performance of an operating offshore wind farm (Mittelmeier et al., 2017) prediction uncertainties need to be reduced. Hansen et al. (2012) studied wake effects at the offshore wind farm Horns Rev in different atmospheric conditions and revealed an influence on the wake magnitude. They also compared turbulence intensities for different stability classes as a function of the wind speed. Below 7 m/s a clear increase in

turbulence intensities can be noticed. Above 7 m/s neutral-unstable conditions are distinguishable from more stable conditions with a constant threshold up to nominal wind speed. Dörenkämper et al. (2014) draw a link from stability via shear to turbulence intensity motivated by the studies of Tambke et al. (2005) and showed its impact on wake effects. Sanz Rodrigo et al. (2015) compared different stability classification methodologies with data from FINO 1 and presented the behavior of shear and turbulence intensity for the proposed atmospheric stability classes. The authors concluded, that in this

particular case TI correlates well for stable cases but at near neutral and unstable cases, shear is supposed to enable better distinction between their proposed nine classes. Atmospheric stability as well as turbulence intensity have been identified as being main drivers for the variation in power production under waked conditions (Dörenkämper et al., 2012; Westerhellweg et al., 2014; Iungo and Porté-Agel, 2014). Therefore state of the art engineering wake models for industrial application like Fuga or FarmFlow are able to take stability effects into account (Özdemir et al., 2013, Ott and Nielsen, 2014).

Stability classification is based on measurements from met masts, buoys or is assisted by remote sensing devices such as Light Detection and Ranging (LiDAR) or Sound Detection and Ranging (SoDAR). For offshore use, these devices are very expensive and therefore often not permanently available. In several studies, LIDAR's have been used to assess the wind speed recovery behind the turbine and wake models have been tuned to match the measured wind speed (Beck et al., 2014, More and Gallacher, 2014).

The purpose of this paper is to investigate wind farm operational data and establish methods of identifying correlations between SCADA statistics and wind turbine wake behaviour caused by different atmospheric conditions.

## 2 Wind farms and measurements

For this investigation, we select three offshore wind farms, i.e. alpha ventus, Nordsee Ost and Ormonde. The first two wind

farms have a well-equipped met mast and provide high quality measurements of hub height wind speed, wind direction, shear and turbulence intensity as well as water temperature.

### 2.1 alpha ventus

The wind farm alpha ventus (AV) is located about 45 km north of the island of Borkum in the North Sea. It consists of twelve turbines of the 5 MW class with a rotor diameter of 126 m and has been commissioned in April 2010. The six

northern turbines (AV1 – AV6) have been manufactured by Senvion. The six turbines in the southern part of the wind farm

are manufactured by Adwen and not considered in this analysis. The FINO1 research met mast is located only 3.2 D (rotor diameters) west of turbine AV4.

The layout of the northern part of alpha ventus (See Fig. 1) allows for investigating the wake behaviour in single and double wake conditions for westerly wind directions. Data was used from 3/2011 to 1/2015. After 1/2015 no data was used, because the installation of the Trianel wind farm in the west is suspected to have changed the environmental conditions of alpha ventus by adding extra turbulence to the inflow.

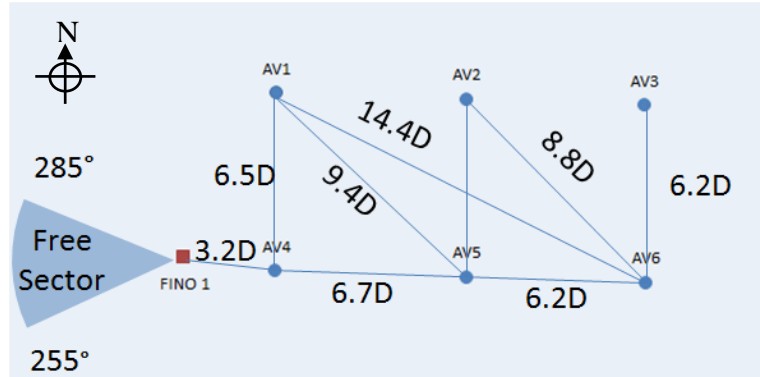

**Figure 1: Scematic Layout of the northern part of alpha ventus wind farm (circles) and FINO1 met mast (red square) with free flow sector and distance in rotor diameters**

## 2.2 Nordsee Ost

The wind farm Nordsee Ost (NO) is located about 35 km north-west of the island of Helgoland in the North Sea. The 48 Senvion turbines have a rated power of 6 MW each and a rotor diameter of 126 m. The met mast is located in the south-western corner of the wind farm (See Fig. 2). In the south, the neighbouring wind farm Meerwind Ost/Süd reduces the sector of free flow for the met mast as well as the possibilities to study multiple wakes higher than triple wake condition without disturbing effects from Meerwind.

The wind farm Nordsee Ost has been fully commissioned in 2015. Data for this analysis is selected from11/2015 – 11/2016. A correlation analysis (described in Sect. 3.2) is performed and the data from a long range LiDAR is analysed. This LiDAR measurement campaign took place within the European Research Project "ClusterDesign".

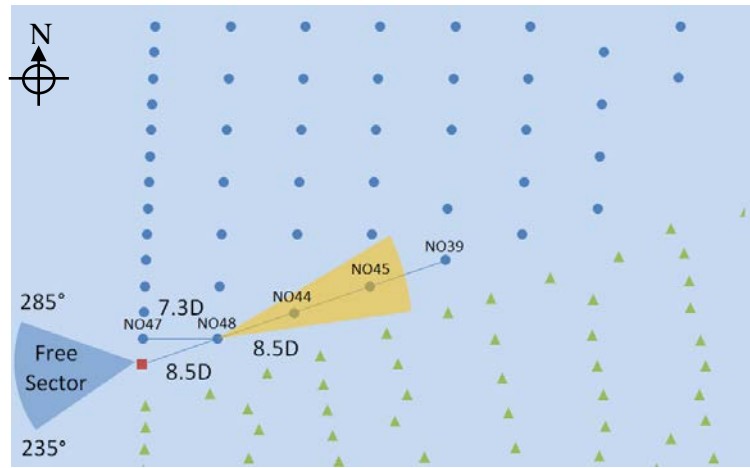

**Figure 2: Nordsee Ost wind farm (blue circles) with neighbouring wind farm Meerwind Süd (green triangles) ,met mast (red square) and distance in rotor diameter (D). Orange area indicates the Plan Position Indicator (PPI) scan from the Windcube 200S, mounted on the helicopter platform of NO48. (Described in Sect. 2.5)**

## 2.3 Ormonde

The Ormonde wind farm consists of 30 Senvion turbines with a rated power of 5 MW each and a rotor diameter of 126 m. The wind farm is located in the Irish Sea 10 km west of the Isle of Walney. The selected data is from 1/2012 – 1/2014. During this period, neighbouring wind farms were operational. Located in the south west are Walney 1 (51 x SWT-3.6-107 Siemens) and Walney 2 (51 x SWT-3.6-120 Siemens), located in the south there is West of Duddon Sands (SWT-3.6-120 Siemens, fully commissioned 30.10.2014) and in the south east there is Barrow (V90 3.0MW Vestas).

The farm layout displayed in Fig. 3 is structured in a regular array which allows for comparing several multiple-wake situations. The inner farm turbine distance for the investigated wake situation from south west is 6.3 D and from north west is 4.3 D.

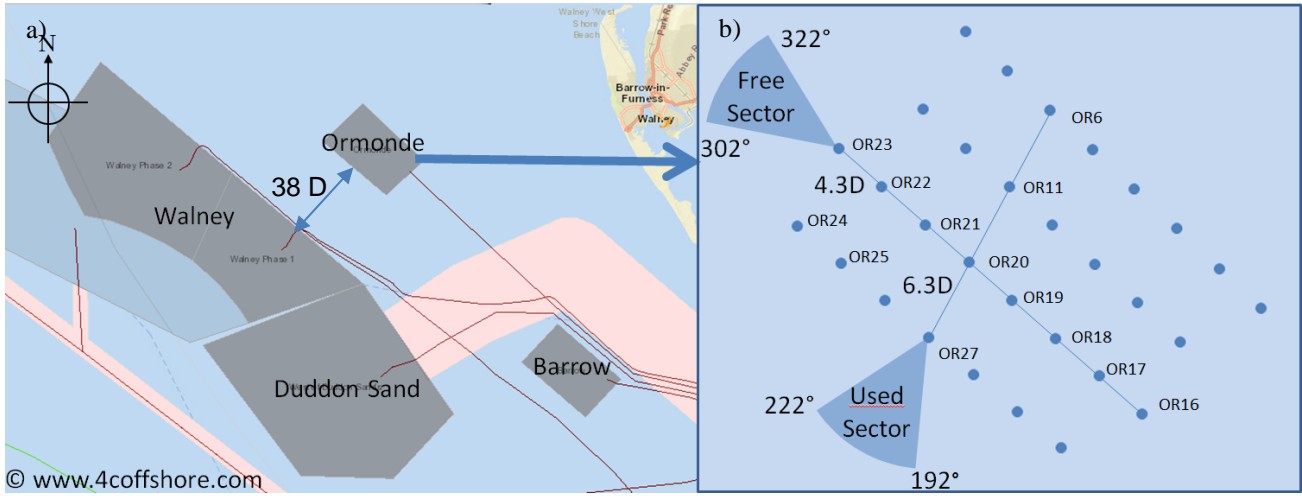

**Figure 3: a) Ormonde and neighbouring wind farms. b) Ormonde wind farm layout with distances in rotor diameters (D) and sectors selected for the analysis.**

## 2.4 SCADA and meteorological data

The SCADA data from all wind farms and the meteorological data consist of 10-min statistics. Each turbine provides wind speed, wind direction, active power, yaw position, and pitch angle. The operational condition of the wind turbine which is used for the correlation with the met mast turbulence intensity is categorized by the minimum active power > 10kW, the maximum pitch angle < 3 ° and the standard deviation of the yaw position < 5 °. These filter criteria's ensure that no stand stills, curtailments or too large yaw activities are included in the data. Furthermore any succeeding 10-min measurement period after a turbine restart is deleted to give the flow enough time to develop. SCADA data are also deleted if the waked turbine produces more power than the free flow turbine. Implausible met mast data are removed and wind directions are corrected for bias by using the orientation of the maximum wake deficit. For the correlation, only sectors of free flow conditions are used. Averages of 30-min water temperature are recorded by buoys at FINO1 and linearly interpolated into the SCADA data.

## 2.5 Long range LiDAR measurements

Within the "ClusterDesign" research project, funded by the European Union, a long-range LiDAR measurement campaign was realized. A Windcube 200S (WLS200S) LiDAR with scan head was placed on the helicopter platform of NO48 (See Fig. 2) from 11/2015 – 5/2016. A differential GPS system composed by three antenna GNSS-Systems of type Trimble SPS855 / SPS555H allows for additional measurements of turbines yaw and nacelle pitch and roll angle. One LiDAR measurement cycle takes about 200s. It includes five Plan Position Indicator (PPI) scans followed by one Range Height Indicator (RHI) scan. Both scans cover a sector of 30° on the horizontal and vertical plane respectively and are centred on

the rotor axis. The scan trajectories have an angular resolution of 1° and measure the wind speed component along the measuring direction every 25 m from 100 m to 2500 m. The LiDAR data is filtered excluding measurements with a poor signal intensity, affected by hard targets and considered outliers.

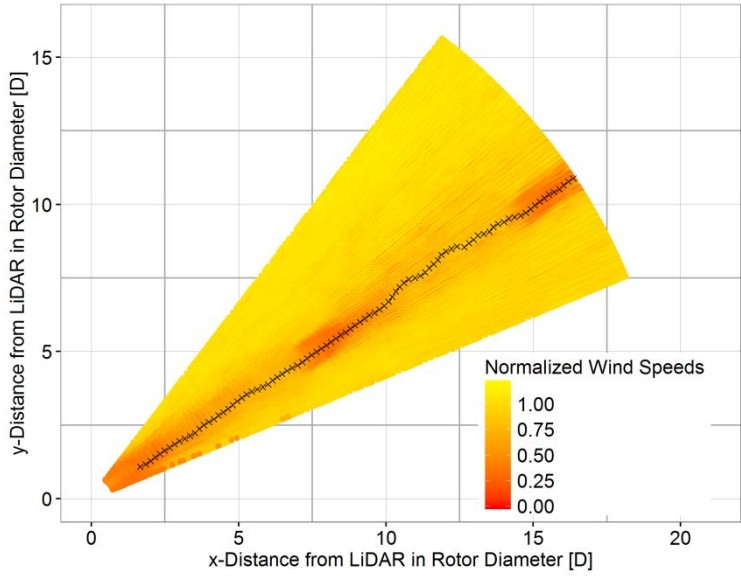

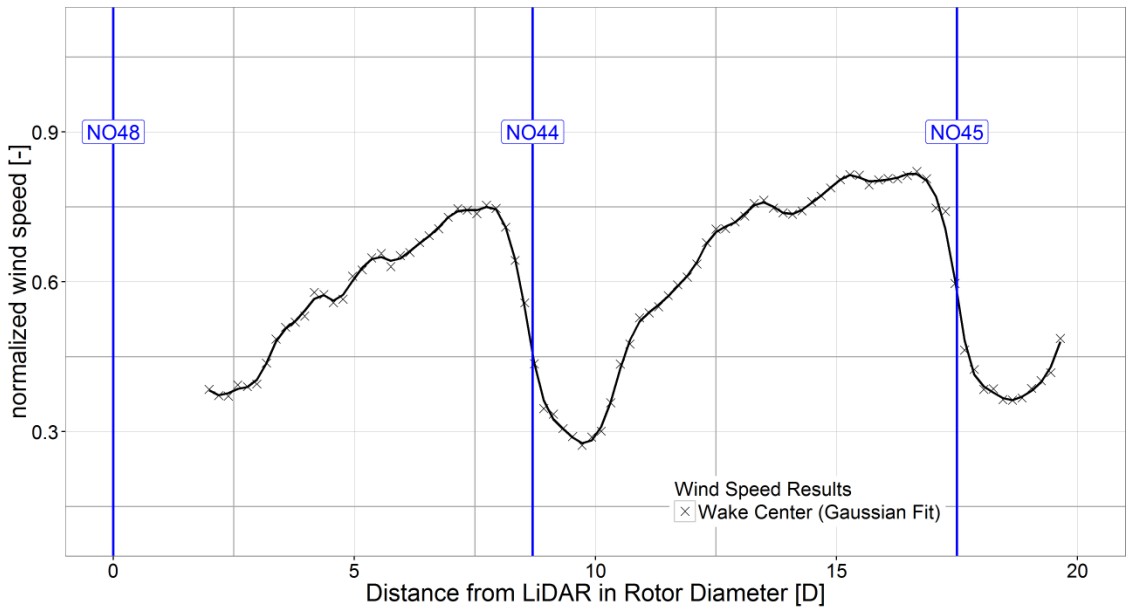

**Figure 4: Upper-plot: Visualization of the horizontal plan position indicator (PPI) scans downstream of NO48. Wind speed is normalized with the inflow wind speed, measured at the met mast. The black crosses are the locations of the wind speed minima's derived from a Gaussian fitting for each measurement distance. Bottom-plot: Normalized wind speed as function of the distance in rotor diameters, extracted from the top-plot for the Gaussian fitted minima's (black crosses).**

The horizontal 10-min average wind speed is calculated on a well-defined grid under the assumption of a negligible vertical component of the wind. The average of the wind component measured by the LiDAR during the considered time interval is included in the region of interest of the addressed grid point. The average 10-min wind direction is provided by the met mast. When the latter measurement is not available, the turbine yaw provided by the differential GPS system is used to estimate the wind direction. A detailed description of the LiDAR data pre-processing can be found in Schneemann et al. (2016).

For the assessment in this paper we are using averages of 10-min periods of horizontal wind speed data evaluated from PPI scans of the wake behind NO48 (See Fig.2). A multiple wake situation can be observed at a wind direction of 237.5 °, when NO45 is in the wake of NO44 and NO48. In Fig. 4 (top) an example for a PPI scan at hub height with averages of 10-min periods of the horizontal wind speed data is displayed. Hard targets like blades and nacelle prevent reasonable wind speed measurement. Wind speeds at these blind regions and between the beam directions are linearly interpolated. The distance from turbine NO48 is normalised by its rotor diameter and the wind speeds are normalized with the corresponding wind speed measured at the met mast. The different colours represent the wind speed relative to the wind speed at the given met mast location. The black crosses display the locations of the estimated wind speed minima for each measured distance equal or greater than 2 D. These minima are supposed to represent the centre of the wake. They are derived at each downstream cross-section with a Gaussian smoothing (Hamilton, 2015) applied to the LiDAR data before fitting a double-Gaussian-type velocity deficit at the near wake (≤2.5 D) and a single-Gaussian-type in the far wake. This distinction prevents an overestimation of the deficit in the near wake where the nacelle still has an influence on the flow shape (Keane et al., 2016). The Gaussian minima (black crosses) do not follow a straight line for the entire scan. This should not be interpreted as meandering as we are looking at averages of 10-min periods.

The lower graph of Fig. 4 shows the resulting, normalized wind speeds over the normalized distance from NO48. The black line with the corresponding black crosses refers to the fitted values of the Gaussian fits.

## 3 Method

Following the objective to propose a signal based on SCADA data that enables to identify the magnitude of wake effect, we first establish stability classification based on Monin-Obukhov surface-layer theory (Monin and Obukhov, 1954) and met mast turbulence intensity. Both approaches can be found in many publications and their influence on the magnitude of wake effects is reported. Secondly, SCADA signals which are effected by turbulence intensity are proposed and the ability of these signals to classify low, medium and high wake effects is analysed.

### 3.1 Stability and turbulence intensity classification

For the determination of atmospheric stability we follow the approach suggested by Ott and Nielsen (2014). Their iterative method is implemented in the software AMOK and derives the inverse Monin-Obukhov-Length $1/L$ from air temperature and wind speed measured at $z\,[m]$ and water surface temperature. Many different thresholds for stability classification have been published and breakdowns from three to nine different classes can be found in the literature (Archer et al., 2016; Sanz Rodrigo et al., 2015). Dörenkämper (2015) and Rajewski et al. (2013) have suggested a classification into unstable, neutral and stable classes (Table 1) based on turbulence intensity (TI) and the dimensionless Monin-Obukhov-Length $\zeta = \frac{z}{L}$.

**Table 1: Definition of stratification with turbulence intensity and the dimensionless Monin-Obukhov-Length $\zeta$**

| Classification | (Dörenkämper, 2015) | (Rajewski et al., 2013) |
|---|---|---|
| Unstable | TI > 6 % | $\zeta < -0.05$ |
| Neutral | 4 % < TI < 6 % | $-0.05 \leq \zeta \leq 0.05$ |
| Stable | TI < 4% | $\zeta > 0.05$ |

Both classifications and their impact on wake effects are compared with FINO1 data in Sect. 4.1 and alpha ventus data in Sect. 4.2.

### 3.2 Correlation analysis

At wind farms with no met mast we have to rely on other signals to describe the differences in power production under different atmospheric conditions. To find the best substitute for a met mast measured turbulence intensity several SCADA signals that are affected by turbulence are correlated to the met mast turbulence intensity which is defined as

$$TI_{mast} = \frac{\sigma_{u_{mast}}}{\bar{u}_{mast}} \,. \tag{1}$$

Analogous to Eq. 1 we define

$$TI_{WT} = \frac{\sigma_{u_{WT}}}{\bar{u}_{WT}} \tag{2}$$

as the turbulence intensity measured with the wind speed anemometer on top of the nacelle with $\bar{u}$ being averages of 10-min periods of horizontal wind speed and its standard deviation $\sigma_u$. Göçmen and Giebel (2016) evaluated 1Hz data from Lillgrund and Horns Rev I and found good turbulence estimators by using a turbine derived "WindEstimate" from look-up tables. When only 10-min statistics are available the signals of interest are the standard deviation of the turbine power

$$PO_{std} = \sigma_P, \tag{3}$$

and the normalisation of this signal with the average power $\bar{P}$. This leads to

$$PO_{TI} = \frac{\sigma_P}{\bar{P}}. \tag{4}$$

Turbulence intensity is dependent on wind speed (Türk and Emeis, 2010). Wake effects are most pronounced at partial load and in this range, shear and $PO_{TI}$ are even more related to wind speed. The relationship between $PO_{TI}$ and the wind speed can be approximated with a third order polynomial,

$$y = \beta_0 + \beta_1 u + \beta_2 u^2 + \beta_3 u^3 + \varepsilon . \tag{5}$$

$y$ is the predicted outcome of the model, $\beta_i$ are constants and $u$ the wind speed. $\varepsilon$ is the residual being normal distributed. Adjusting $PO_{TI}$ with this model

$$PO_{TI\,norm}(u) = \frac{PO_{TI}}{\beta_0 + \beta_1 u + \beta_2 u^2 + \beta_3 u^3} \tag{6}$$

5    enables to select constant thresholds. For the wind speed range in partial load, the polynomial is always greater than zero.

(At AV4 the constants of the third order polynomial are $\beta_0 = 99.9$, $\beta_1 = -32.5$, $\beta_2 = 3.9$ and $\beta_3 = -0.16$. See Sect. 4.4)

### 3.3 Classification of wake magnitude

In this Section, we describe the methodology to establish thresholds for sub-setting the measurements into weak, medium
10    and strong wake effects. The thresholds are estimated with a three-step approach. At first we select the normalized power (waked turbine, normalized by the power of a free flow turbine) for a small sector (10°) in the full wake for the relevant wind speed range (8 ± 1 m/s) (Fig 5a). Secondly we eliminate the dependency on wind direction by subtracting from the normalized Power $P_n$ its mean value of each wind direction bin (binwidth = 2°) and obtain $P_{nn}$ (See Fig. 5b).

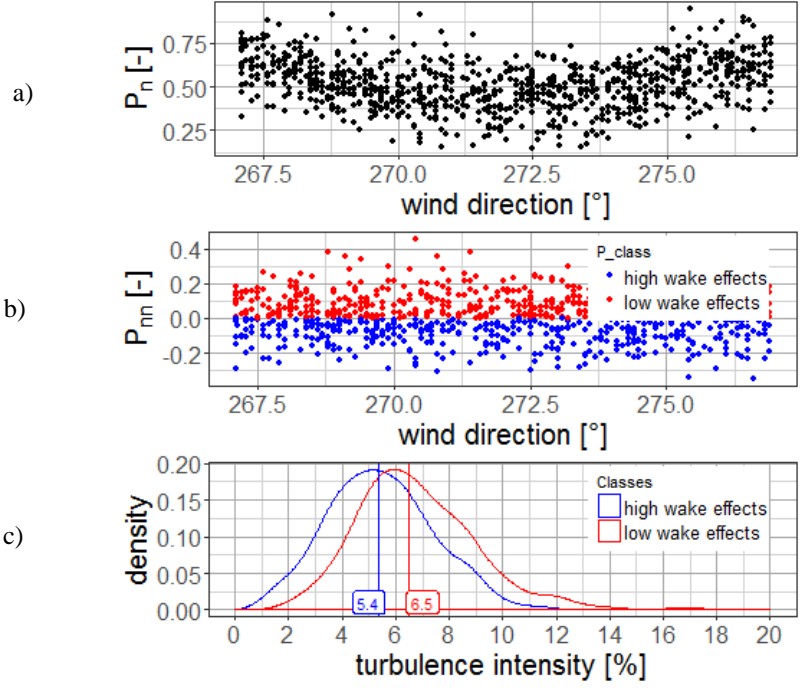

Figure 5: a) Power of AV5 normalized with the power of AV4 for 10° sector around the full wake for wind speed range 8 ± 1 m/s.
b) Wind direction dependency corrected normalized power. Each 2° bin is normalized with its bin average. Data is classified in
high and low wake effects. c) Density distribution of the turbulence intensity from the met mast for each data subset with median.

The third step divides the data set into high wake effects (values < 0) and low wake effects (values >= 0). The median of each data subset is proposed to allocate the thresholds (See Fig. 5c).

## 4 Results and Discussion

### 4.1 Stability and turbulence intensity classification

5     For the selected period at alpha ventus, Fig. 6 shows the stability distribution with the proposed thresholds of $\zeta = 33/L$ from Table 1. There is a clear tendency of having more stable conditions with increasing wind speed. This trend is also visible with turbulence intensity measured at FINO1. The strong wind speed dependency of $PO_{TI}$ leads to an overestimation of this behaviour. The proposed normalization as described in Eq. (6) reduces this information.

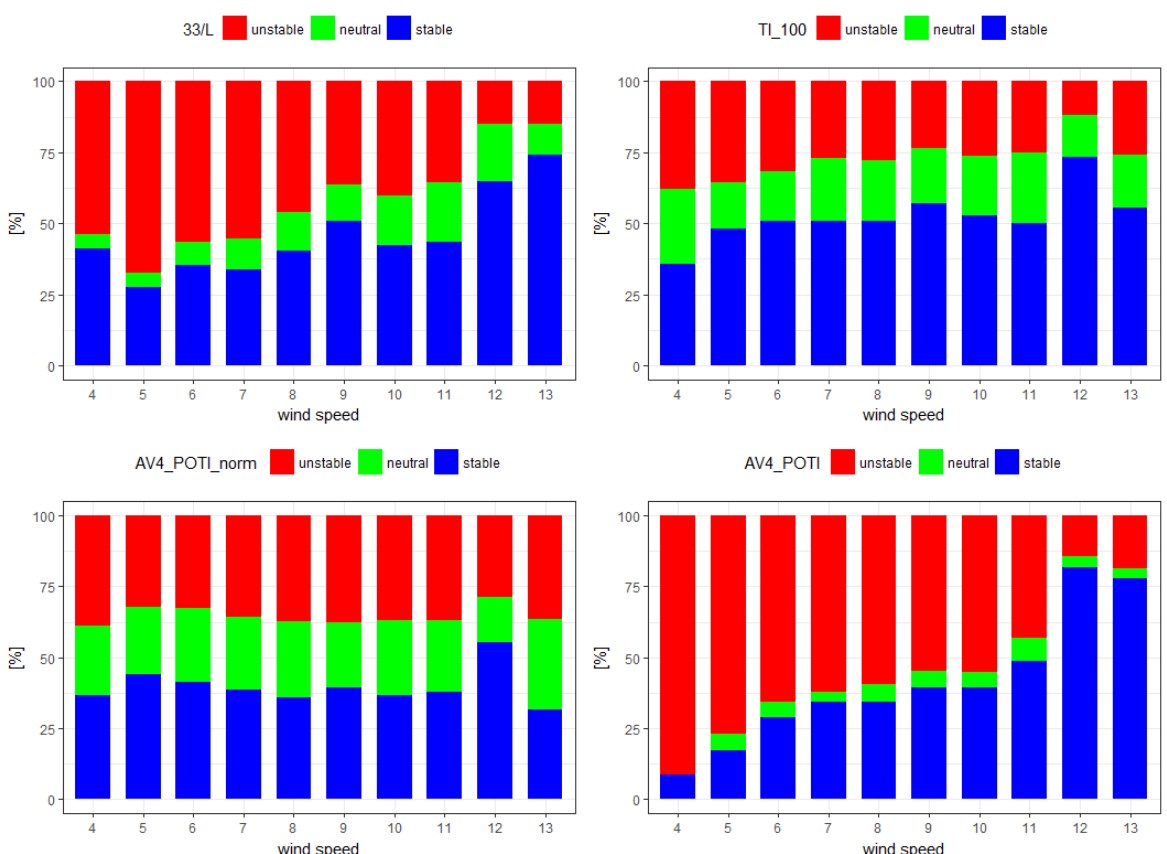

10     **Figure 6: Distribution of stability based on $\zeta$ as a function of wind speed for different parameters of interest.**

Figure 7 presents the data from FINO1 with bin averaged met mast turbulence intensity measured at 100 m (TI_100), turbulence intensity from the turbines nacelle anemometer at hub height (AV4_TI), power law coefficient from 40 m and

90 m heights (alpha_40_90) and the standard deviation of the power divided by the average power (AV4_POTI). Each signal is plotted as function of the wind speed for the three proposed stability classes based on $\zeta = \frac{33}{L}$.

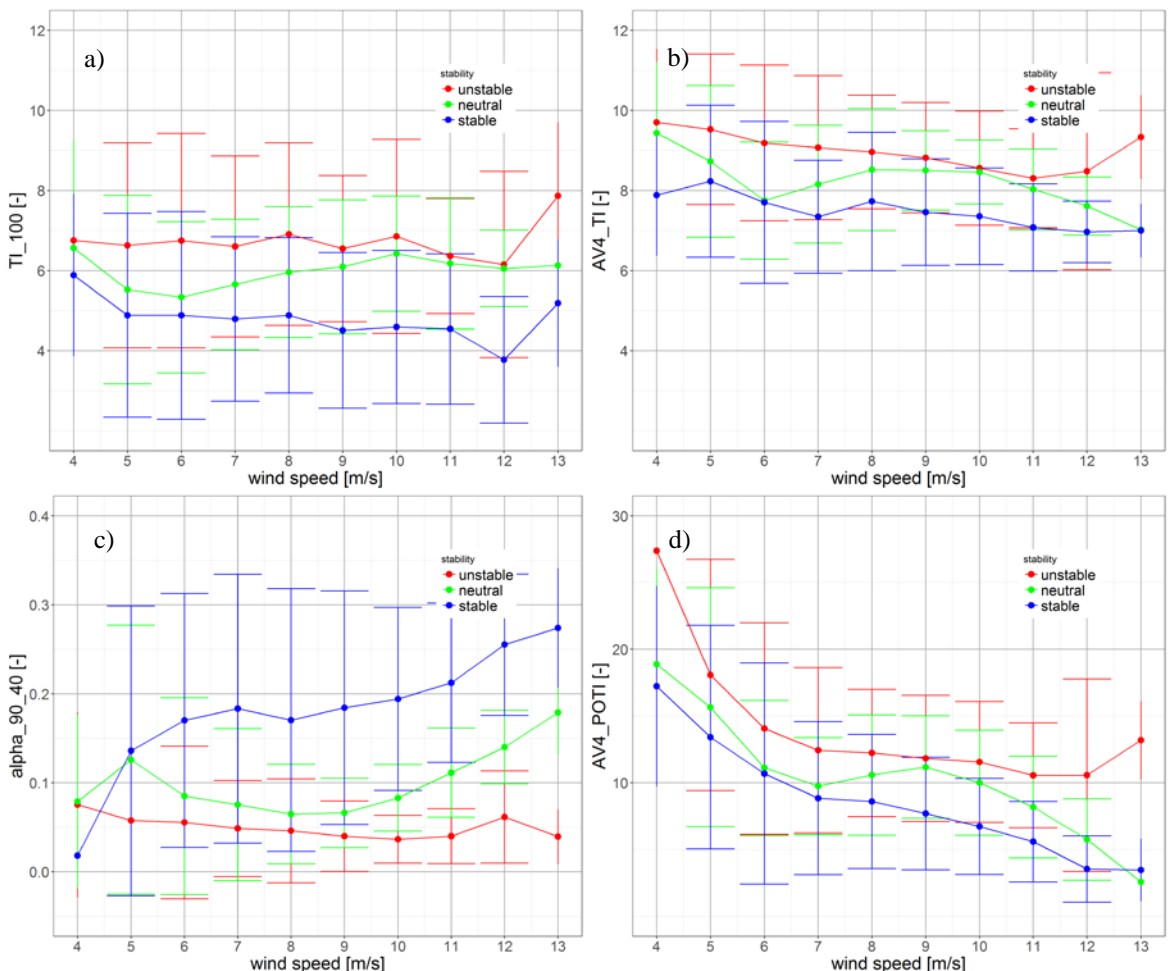

Figure 7: a) Turbulence intensity at FINO1, b) Turbulence intensity at AV4, c) Power Law coefficient for shear at FINO1 hights at 40m and 90m and d) Standard deviation of the power divided by the average power for free flow conditions at AV4. All four measurements are provided as function a of the wind speed for three stability classes based on $\zeta = \frac{33}{L}$.

For all signals in Fig. 7 the differences between the stability classes is visible. Whereas shear (See Fig. 7c) has problems in light winds, it gives the best results for the higher wind speeds. Turbulence intensity (See Fig. 7a) is the most constant signal for the selected wind speed range. The turbulence intensity measured at the nacelle (See Fig. 7b) is much higher due to the location of the anemometer behind the rotor. This comprises the variation and weakens the ability to distinguish between the stability classes. The relationship between the standard deviation of the power divided by the average power (See Fig. 7d) and the wind speed is very dominant and can be approximated with a third order polynomial.

## 4.2 Impact on power production

Alpha ventus data from almost four years of operation is used to evaluate the influence of atmospheric stability and turbulence intensity on the wake development. Figure 8 shows the different wake behaviour under different turbulence conditions. The top row of plots are single wake condition of turbine AV5 in the wake of AV4. The second row displays the

same evaluation but for the double wake condition of AV6 in the wake of AV4 and AV5. The left side is a normalised power deficit as function of the wind direction for a wind speed range from 7 m/s to 9 m/s. On the right side, there is the normalised power as function of the wind speed for a sector width of 10 °. Each graph states the total number $N$ of data points which have been split into stable (blue dots), neutral (green diamonds) and unstable (red triangles) data sets. Each symbol is the average of a 2 ° bin (2 m/s bin) and the error bars indicate the standard error of the mean.

For the single wake, a clear distinguishable difference between the stable and unstable power deficit is visible. The largest deviation is found in the full wake. The second wake has a less pronounced difference in power which can be explained by the fact, that the first turbine operating in the wake supports the mixing with the ambient wind speed. Another interesting effect is noticeable in the top left plot. The difference in power for the different stabilities is higher at the right hand side of the deficit in downstream direction. This right drift of the wake in stable conditions has also been observed in LES

simulations by Vollmer et al.(2016). This effect is even more pronounced when the data is distinguished with $\zeta = 33/L$ (See Fig. 9). The total magnitude of wake effects is better distinguished by the classification with turbulence intensity. For this reason we will consider from now on turbulence intensity as reference.

The turbulence intensity for this classification has been measured at 100 m which is the largest height at the FINO1 met mast. The second height of the FINO1 met mast (90 m) is closer to hub height (92 m), but the strong mast structure and the

boom orientation of 135 ° cause disturbance for wind directions within the selected sector for our investigation. No further correction, e.g. to account for the difference in height was necessary according to the findings of Tuerk (2008).

For the wind farm Nordsee Ost (NO) only one full year of SCADA data and six month of LiDAR data is available for this investigation. Eighteen PPI scans (as described in Sect. 2.5) with full wake conditions are available and categorized

according to the classification in Table 1. Fig. 10 displays the wind speed measured with the LiDAR normalized with the inflow wind speed measured at the met mast. The wind speed recovers faster for the unstable turbulence class than for the neutral and stable class. The second and third turbine in the row for the investigated wind direction are marked with blue vertical lines. The decreased wind speed in the induction zone in front of each downstream turbine is clearly visible. The error bars indicate the standard error of the mean. For the stable class $TI \leq 4\%$ only one set of PPI scans is available.

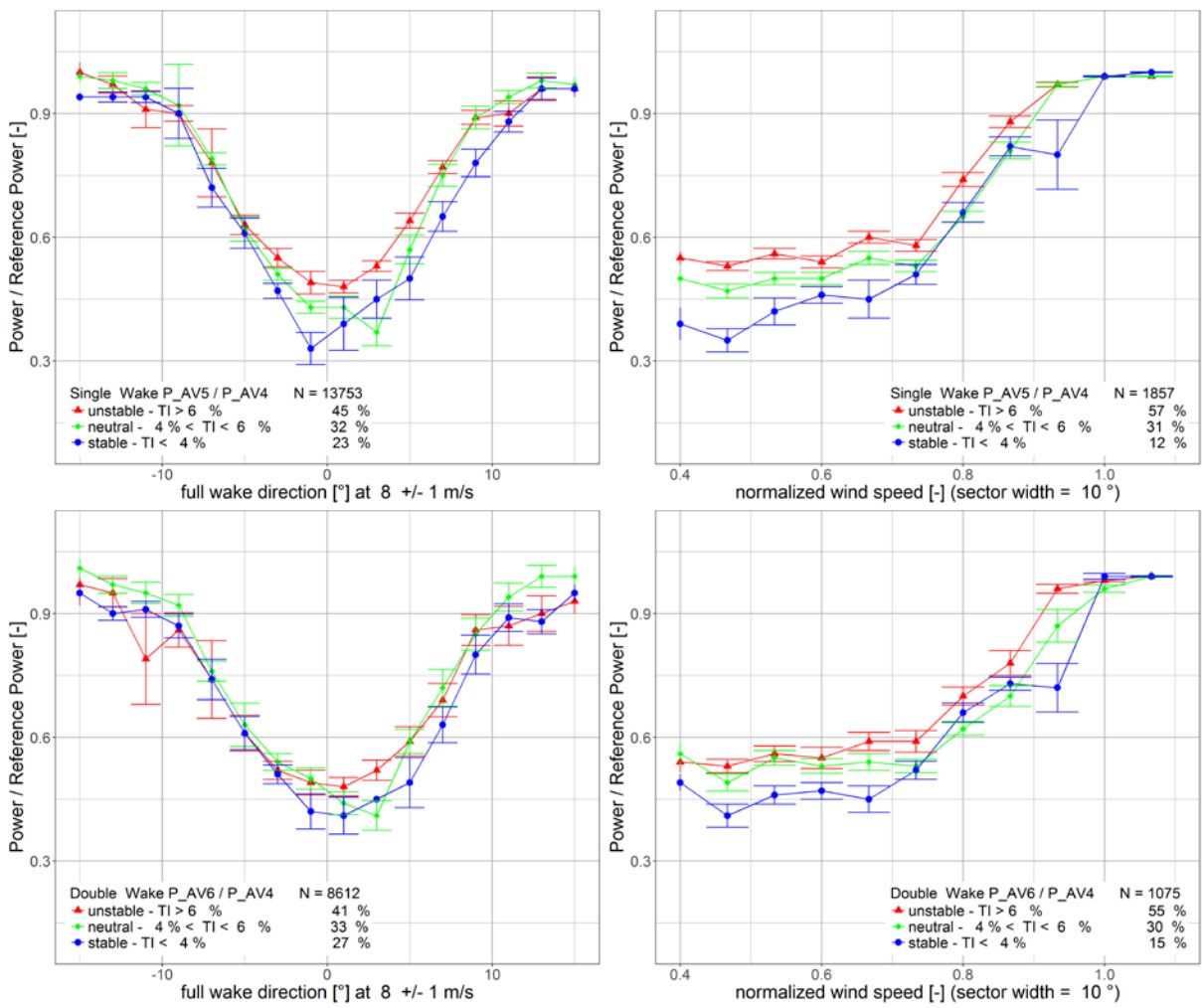

**Figure 8: Wake effects in alpha ventus (AV) under different atmospheric conditions classified by met mast turbulence intensity. Power of downstream turbine normalised with free flow turbine. Upper row: single wake, bottom row: double wake. Left column: Normalised Power as function of wind direction, right column: Normalised power as function of wind speed.**

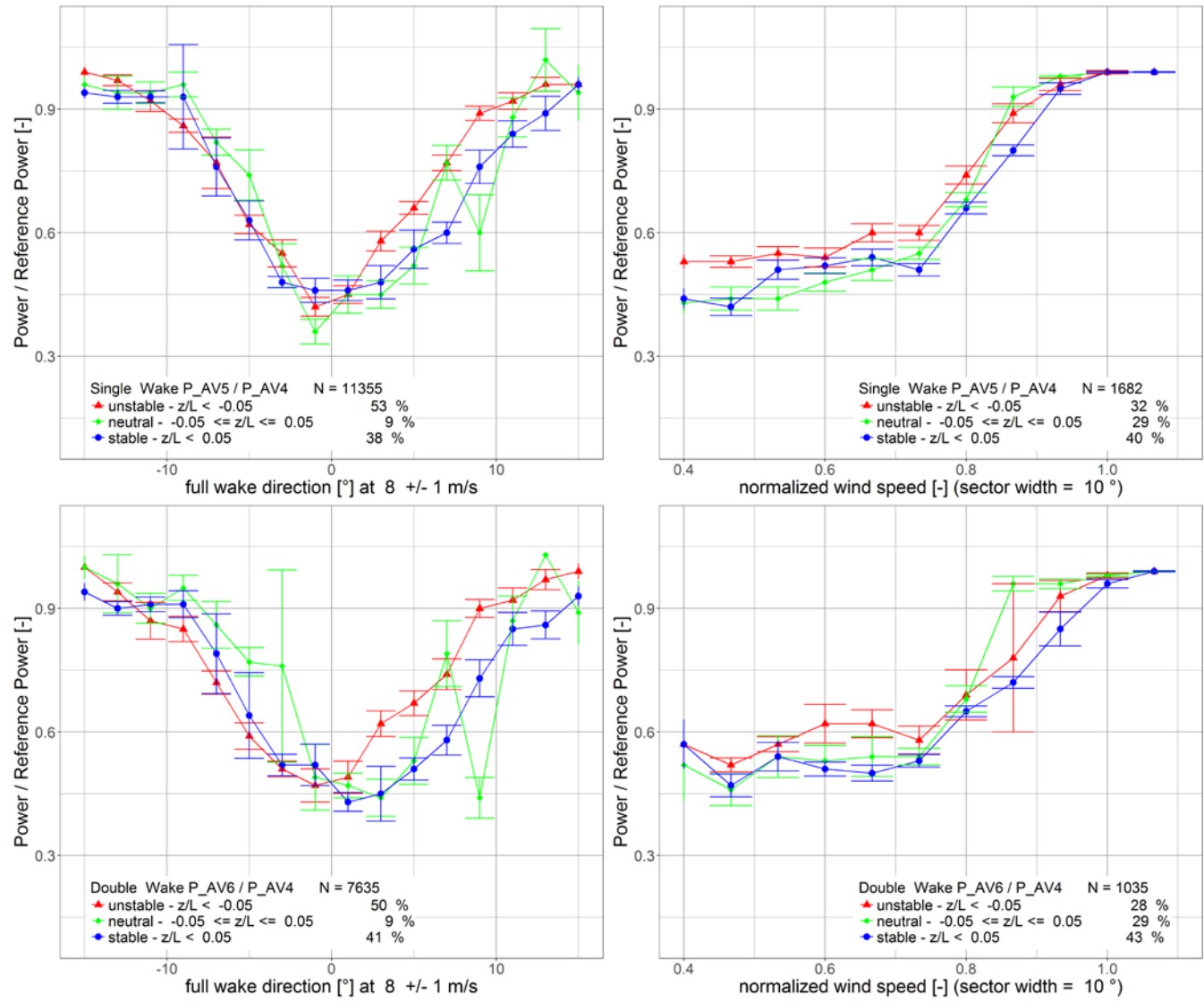

**Figure 9: Wake effects in alpha ventus (AV) under different atmospheric conditions classified by $\zeta = 33/L$. Power of downstream turbine normalised with free flow turbine. Upper row: single wake, bottom row: double wake. Left column: Normalised Power as function of wind direction, right column: Normalised power as function of wind speed.**

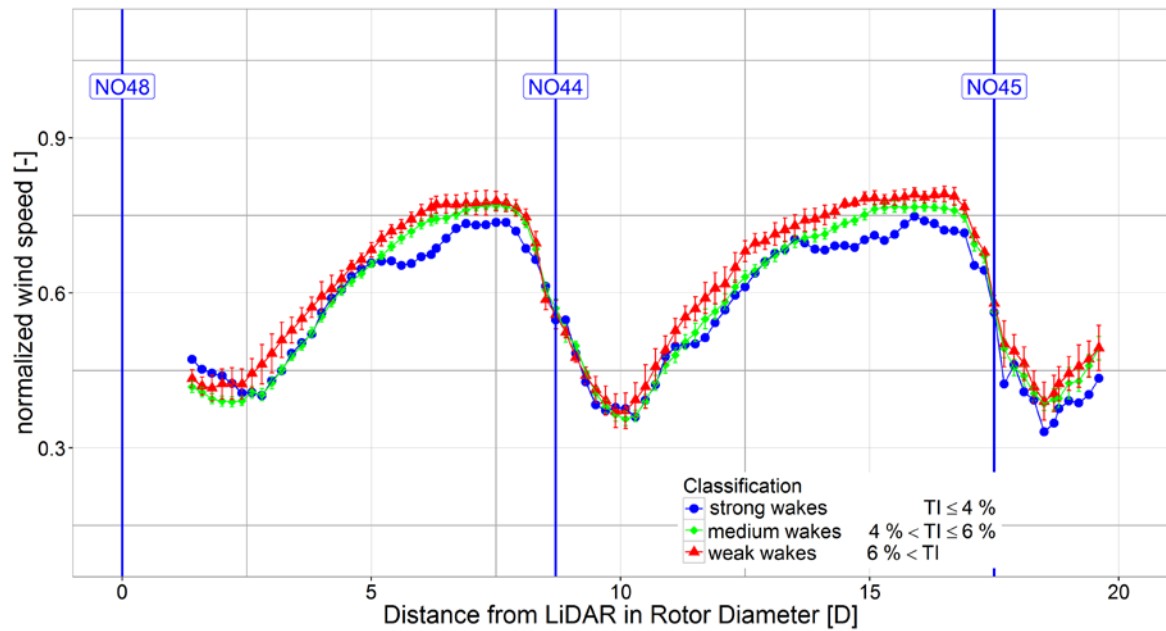

**Figure 10: Wind speed recovery at wake centre on hub height behind NO48 for different turbulence stability classes. The wind speed is normalized with the inflow wind speed and the distance from the LiDAR on NO48 downstream is displayed in multiples of rotor diameters**

### 4.3 Correlation analysis

In the next step, we correlate the SCADA signals described in Sect. 3.2 with the turbulence intensity measured at the mast. In Fig. 7 a panel plot is displayed. The graphs on the diagonal present the histogram and density distribution for the respective variable. The panels above the diagonal provide the Pearson correlation coefficients. The lower panels are scatter plots for the two variables with a fitted linear regression line. The colours of the points indicate the three stability classifications (blue: stable, green: neutral, red: unstable) determined with the met mast turbulence intensity.

The correlation between met mast and turbine TI in subplot (1 , 2) equals to 0.55. This poor result can be explained by the nacelle wind speed measurement position behind the rotor, which induces additional disturbance to the flow.

The highest correlation with the met mast TI is obtained with the standard deviation of the turbine power divided by its average active power ($PO_{TI,AV4}$) in subplot (1 , 5). Although a correlation of 0.66 is not perfect, it is still better than the turbulence measured with the nacelle cup anemometer. Especially in the low turbulence region, the scatter plot proves to be denser. Very similar results are obtained when applying the same analysis to AV1 and AV2.

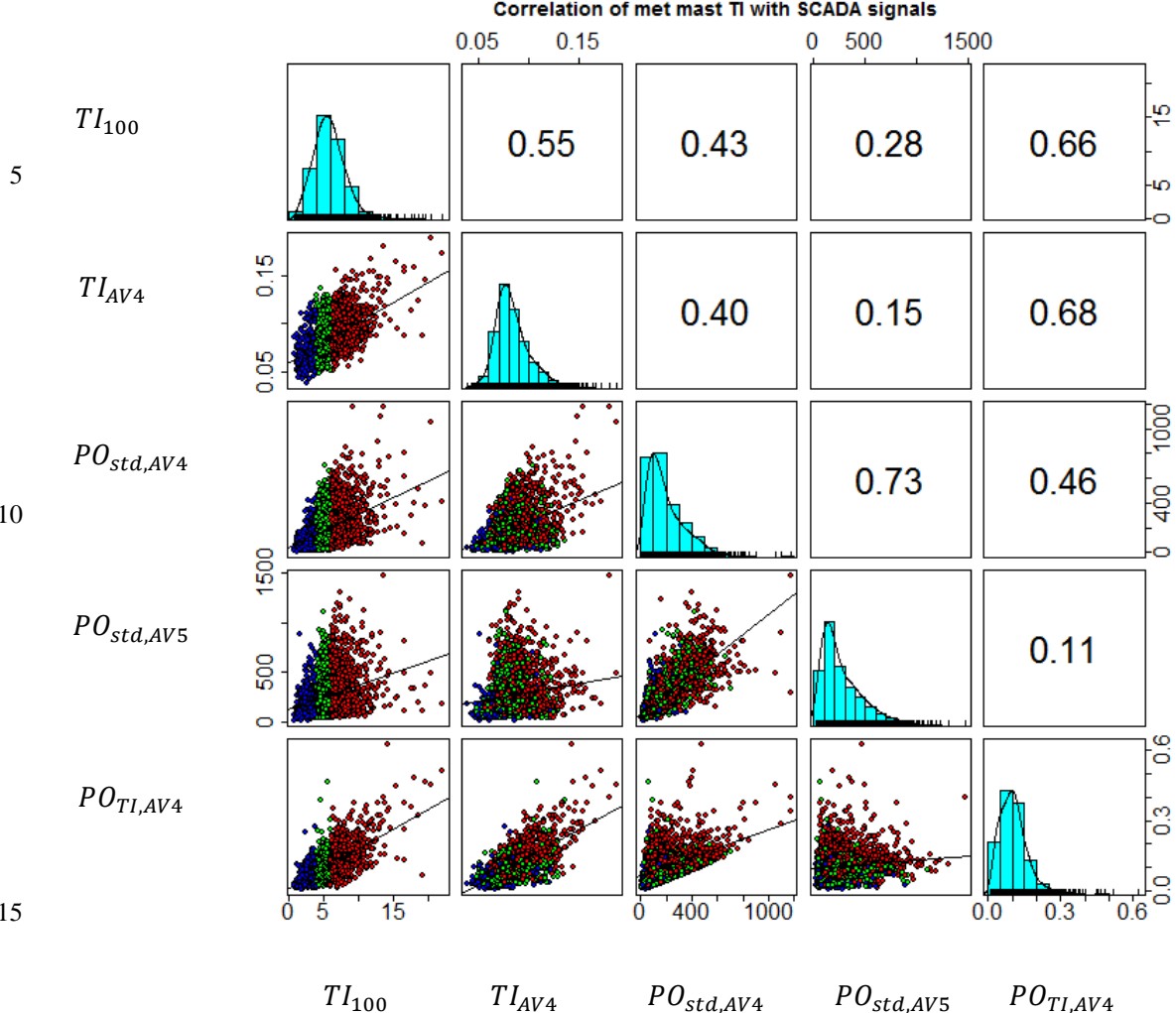

**Figure 11: Correlation matrix.** Turbulence intensity from met mast ($TI_{100}$) is correlated with the TI measured with the nacelle anemometer of AV4 ($TI_{AV4}$), the standard deviation of the 10 min power of AV4 ($PO_{std,AV4}$), the standard deviation of the 10min power of AV5 ($PO_{std,AV5}$) and the standard deviation of the power divided by the average power of AV4 ($PO_{TI,AV4}$). All dimensions are in [%] except for the standard deviation of the power which is in [kW].

To check the validity of these results, we use data from Nordsee Ost (NO). Figure 12 provides the information corresponding to Fig. 11 but for a different turbine type, met mast and a different location in the North Sea.

The correlation reveals the best result for the $PO_{TI,NO47}$ signal (0.62). $TI_{NO47}$ derived from the nacelle cup anemometer gives 0.60 . The difference between these two signals is much smaller than in alpha ventus. A different blade design and the distinct turbine nacelle met mast layout might be the reason for this.

Both correlation analyses show that the new artificial SCADA signal, derived from the standard deviation of the power divided by its average active power $PO_{TI}$ is the most suitable among the selected signals to substitute a met mast $TI_{mast}$ for our purpose. In the next Section, we check the influence of this new signal on the estimated power production in the wake.

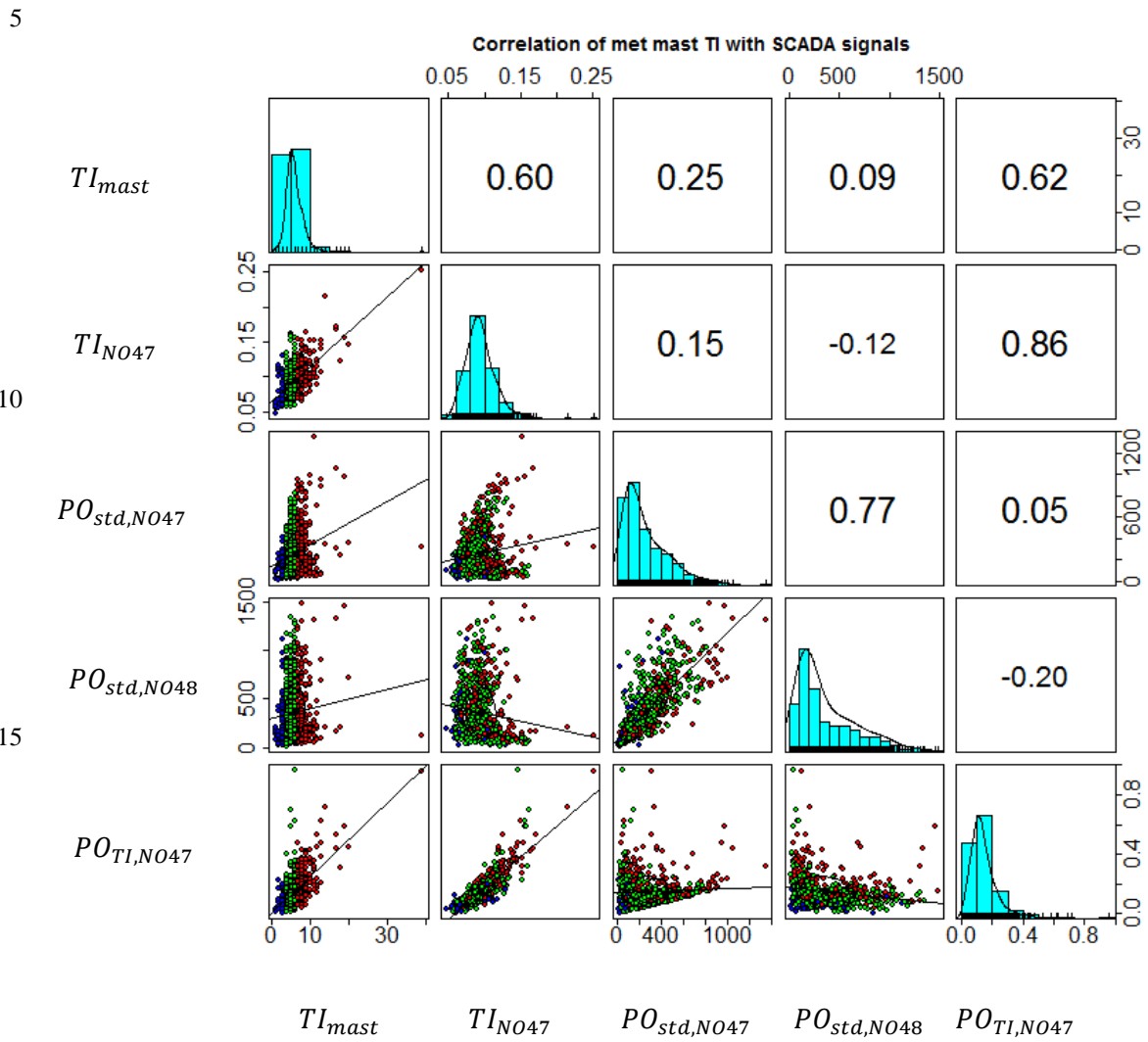

**Figure 12: Correlation analysis for Nordsee Ost. Turbulence intensity ($TI_{mast}$) measured at hub height is correlated with the TI measured with the nacelle anemometer of NO47 ($TI_{NO47}$), the rotor estimated wind speed ($TI_{NO47_{ues}}$), the standard deviation of the 10 min power of NO47 ($PO_{std,NO47}$), the standard deviation of the 10 min power of NO48 ($PO_{std,NO48}$), the standard deviation of the power divided by the average power of NO47 ($PO_{TI,NO47}$). All dimensions are in [%]except for the standard deviation of the power which is in [kW].**

## 4.4 New classification and validation

In Sect. 4.2 we demonstrated the correlation of the SCADA signal $PO_{TI}$ with the turbulence intensity measured at a met mast in free flow conditions. In Fig. 7d a strong wind speed dependency for the range of interest prevents a constant threshold establishment. Therefore the adjustment with Eq. (6) is proposed. Figure 13 shows $PO_{TI_{norm}}$ as a function of wind speed for the three stability classes based on $\zeta$. The error bars are 1 standard deviation.

For AV4 the constants of the third order polynomial are $\beta_0 = 99.9$, $\beta_1 = -32.5$, $\beta_2 = 3.9$ and $\beta_3 = -0.16$.

.

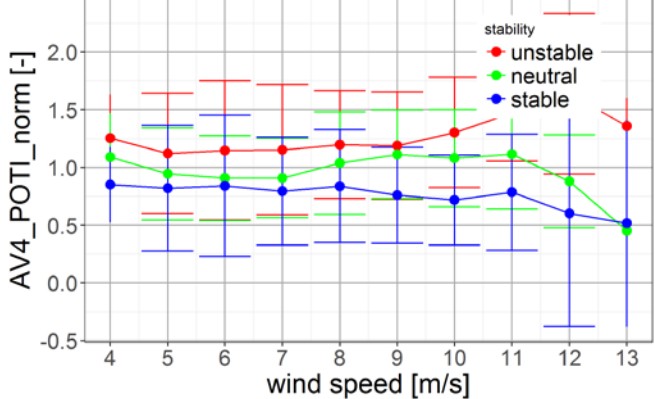

**Figure 13: Normalized SCADA signal ($PO_{TI_{norm}}$) for the classification of the magnitude of wake effect.**

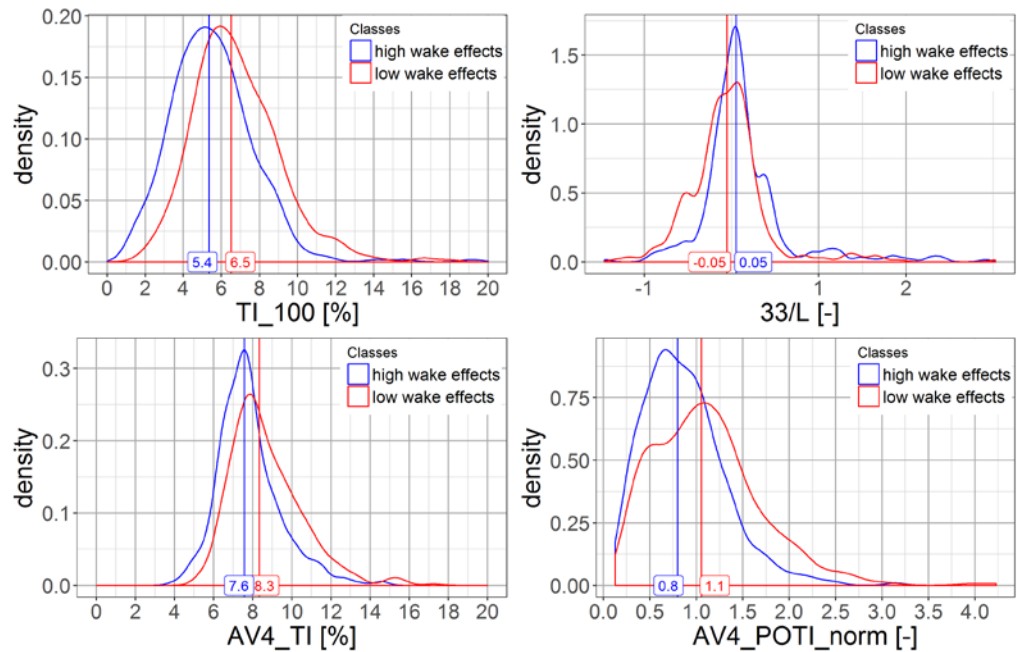

**Figure 14: The thresholds are displayed as vertical lines representing the median of the density distribution for the signal of interest.**

With the methodology from Sect. 3.3 we obtain different density distributions for high and low wake effects (See Fig. 14). The thresholds are derived from the median of the data distribution. For $\zeta$ we obtain exactly the same values as proposed in Table1.

Comparing the two turbulence intensities in Fig. 14 a) and c) one can clearly see the increased turbulence behind the rotor especially for low TI values. The distribution for the nacelle measurement is compressed in a way, that the low measurements have up to 4% difference but the high turbulence intensities are more or less comparable. This effect reduces the ability of the nacelle turbulence intensity to distinguish between the wake magnitude. A more clear separation between the wake classes can be achieved with the new proposed $PO_{TI_{norm}}$.

The quality of the established relationship in terms of dependency on turbine type, layout and location of the wind farm is tested by applying the same classification on a different wind farm where no met mast is available.

In the next Sections, the ability of this new signal to distinguish between different environmental stratification is analysed. Table 2 shows the proposed thresholds for the different parameters under investigation.

**Table 2: Summary of thresholds for the different classes of interest at alpha ventus:**

| Category | $\zeta = z/L$ [-] | $TI_{100}$ [%] | $AV4_{TI}$ [%] | $AV4\_POTI_{norm}$ [-] |
|---|---|---|---|---|
| Weak wakes | $\zeta < -0.05$ | $TI_{100} > 6.5$ | $AV4_{TI} > 6.5$ | $POTI_{norm} < 0.8$ |
| Medium wakes | $-0.05 \leq \zeta \leq 0.05$ | $5.4 \leq T_{100} \leq 6.5$ | $5.4 \leq AV4_{TI} \leq 6.5$ | $0.8 \leq POTI_{norm} \leq 1.1$ |
| Strong wakes | $\zeta > 0.05$ | $T_{100} < 5.4$ | $AV4_{TI} < 5.4$ | $POTI_{norm} > 1.1$ |

### 4.4.1 alpha ventus (AV)

The classification of wake effects by the $PO_{TI_{norm}}$ signal is illustrated in Fig. 15 analogous to Fig. 8 where the turbulence intensity $TI_{mast}$ is used.

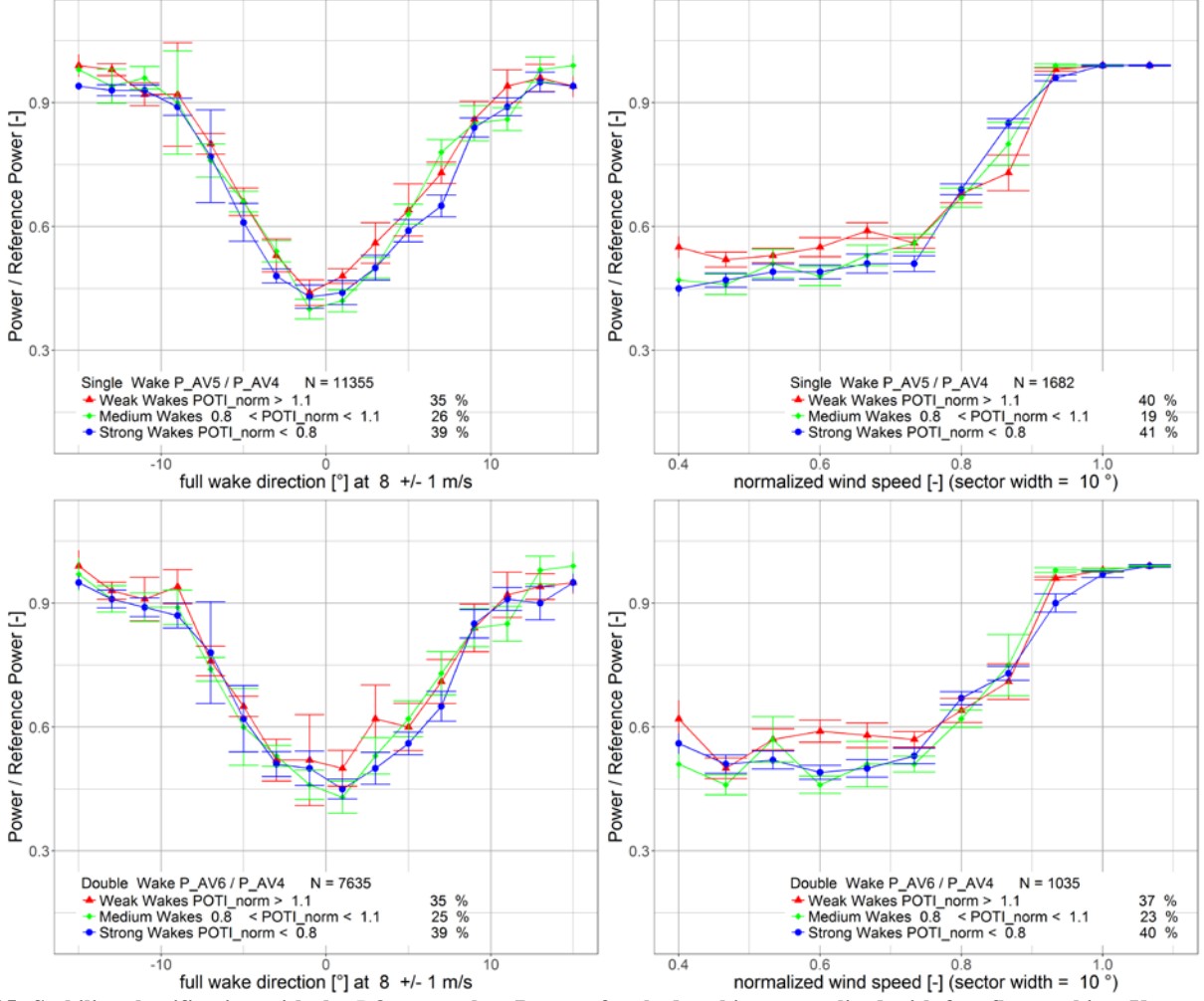

**Figure 15: Stability classification with the $PO_{TI_{norm}}$ value. Power of waked turbine normalised with free flow turbine. Upper row: single wake, bottom row: double wake. Left column: Normalised Power as function of wind direction, right column: Normalised power as function of wind speed.**

A clear difference in power production between stable and unstable cases can be identified in the single wake. The differences in double wake are again less pronounced. Compared to the TI classification, the curves for the neutral case are not as clear as in-between the stable and unstable curves and in the normalized power curve plots (right column) the stable conditions can only be highlighted up to the wind speed of rated power for the free flow turbine. This can be explained with the fact, that at rated power the pitch controller rather than the power variation is governing the turbine reaction on turbulence intensity. This leads in Eq. (4) to a significant decrease of the numerator and keeps the denominator constant.

### 4.4.2 Nordsee Ost (NO)

The classification of wake effects in Fig. 16 is based on $PO_{TI\_norm}$ and analogous to Fig. 10 where turbulence intensity $TI_{mast}$ is used. For these plots the Gaussian fitted minima's of the normalized wind speed (wake centres) measured by the LiDAR are plotted for each distance behind NO48.

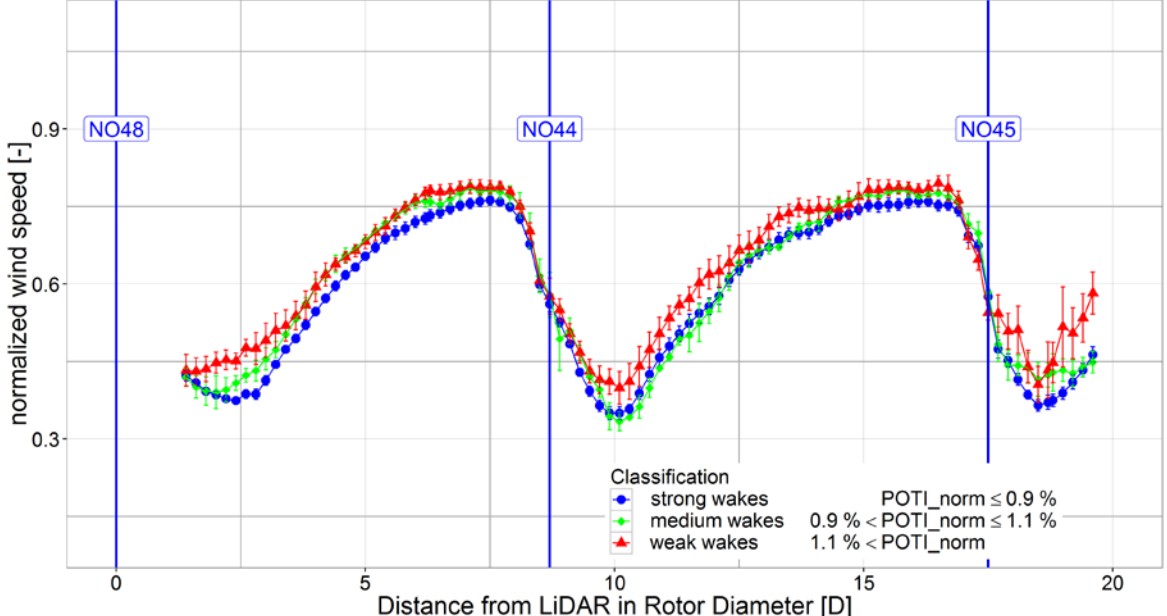

**Figure 16: Wind speed recovery behind NO48 for different $PO_{TI_{norm}}$ classes. The wind speed is normalized with the inflow wind speed and the distance from the LiDAR on NO48 downstream is displayed in multiples of rotor diameters.**

This result states the ability of $PO_{TI_{norm}}$ to distinguish between different wake effects. The single wake has a less pronounced difference between the three classes and the slope of the wind speed recovery is smaller than the double wake case. E.g. 5D behind the first turbine, the wind speed has recovered to approximately 70% of the free flow wind speed and in the second wake 5D behind NO44 we see already more than 75%. This fact leads to the performance increase at the third turbine (NO45) compared to the second turbine (NO44). Wake added turbulence of NO48 is helping to recover the wind speed.

### 4.4.3 Ormonde (OR)

Finally the transferability of classification boundaries to other wind farms where no met mast is available is of interest.
First we have a look at the sensitivity of the signal $PO_{TI}$ in terms of turbulence from neighbouring turbines and wind farms. In Fig. 17, the directional bin averaged $PO_{TI}$ from OR24 is able to identify the location of its neighbours. The magnitude

allows to determine which turbine is next (OR25 at 4.2D) and which is further away (OR22 at 6.7D, OR23 at 6.6D and OR21 at 9.1D).

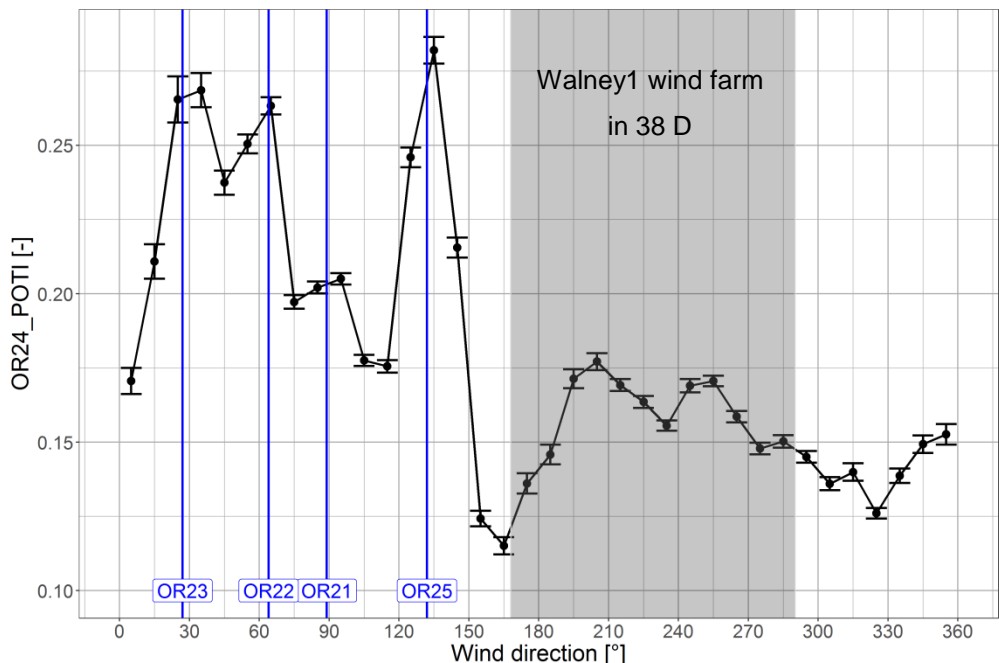

**Figure 17: The signal $PO_{TI}$ is sensitive to wake-induced turbulence from neighbouring turbines and wind farms. With $PO_{TI}$ from turbine OR24 it is even possible to rank the distance of the neighbours being OR25 the closest with 4.2D and OR21 the farthest with 9.1D.**

The grey area represents the geometrical location of the neighbouring wind farms Walney 1 and 2. The closest distance to OR24 has Walney 1 with approximately 38.8D (SWT-3.6-107 Siemens). The two peaks at 208° and 255° are wind directions for which multiple turbines of the neighbouring wind farm are aligned in a clear row of full wake situations. The increase from 345° to 15° can be explained with the coastline that gets quickly closer in clockwise direction.

Secondly we have a look at the influence on the wake recovery. With south westerly wind direction, we focus on single wake, double wake and triple wake conditions behind turbine number OR27 for a sector of 10° around the full wake situation. And for north westerly wind directions we investigate the rows of turbines behind OR23. The main differences between these two directions are the average level of inflow turbulence intensity and the different spacing between the turbines. In Fig. 17, the inflow turbulence level from north west (sector of 302 ° to 322 °) is much lower (bin average $\overline{PO_{TI}} \approx 12.5$ %) than from south west (sector of 192 ° to 222 °, bin average $\overline{PO_{TI}} \approx 17.5$ %) due to the wake effects from Walney 1 in more than 38D distance.

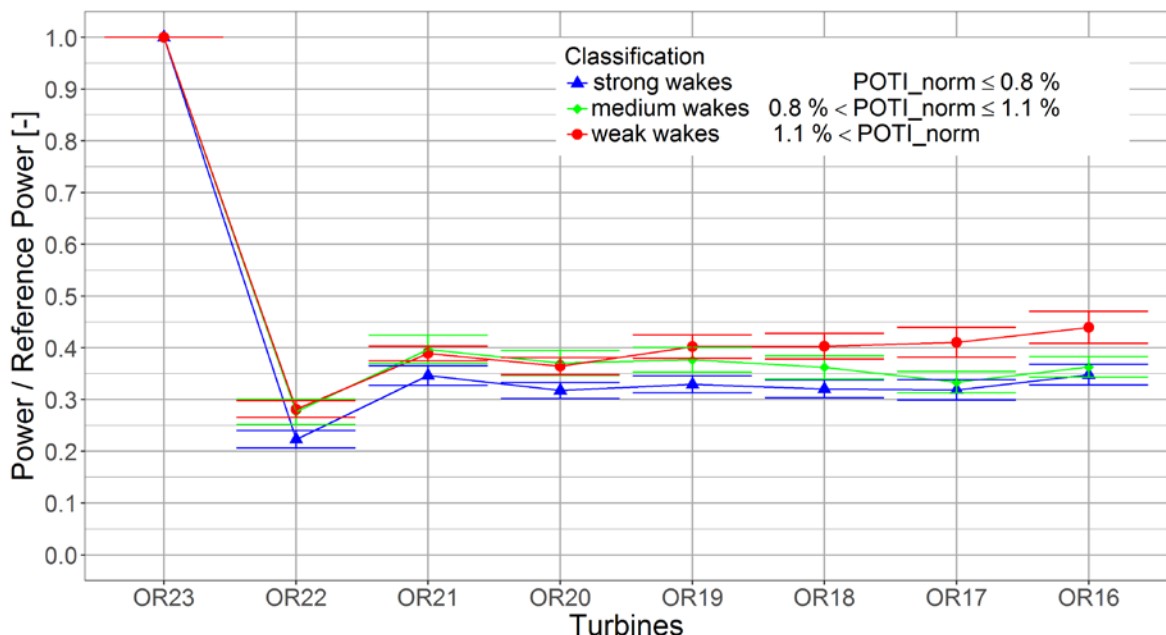

**Figure 18: Normalized power for each turbine along the row behind OR23 for a wind direction of 312 ° and a 10 ° sector width and 8 m/s ±1. Stability classes distinguished with the signal $PO_{TI_{norm}}$ from OR23.**

For north westerly wind direction, Fig. 18 provides a view on different wake effects for the proposed classification. The normalized power for each turbine in the row behind OR23 is displayed (wind from left to right). Wind speed is filtered for $8 ± 1$ m/s and the wind direction is 316 ° with a sector width of 10 °. The largest wake effects are detected at OR22. This underlines the observation from the LiDAR measurements in NO. The first wake is the strongest and all consecutive wakes are better mixed due to wake added turbulence. The difference in power production between high and low turbulence conditions is in the range of 10%, which also demonstrates the importance of this effect for wake model developers to take it into account.

The south westerly wind direction is analyzed in Fig. 19, which is a similar illustration as Fig. 8 and Fig. 9. It is still possible to identify different wake behaviour for the different classes but the effect is less pronounced than in the previous examples. A higher level of inflow turbulence intensity contributes to the mixing of the wake with free wind. Hence at lower inflow turbulence levels the effect of the wake added turbulence is larger.

Further investigations are necessary to account for controller properties and to fill the normalized wind speed range [0.75 – 1], beyond the rated wind speed of the turbine in free flow conditions.

In performance monitoring of offshore wind farms the newly aggregated SCADA signals can be used as an auxiliary quantity to classify different atmospheric conditions. Advanced engineering wake models which are able to take turbulence intensity or stability parameters into account, may be parameterized by these artificial turbine signals in order to improve their prediction of wind turbine power production under waked conditions.

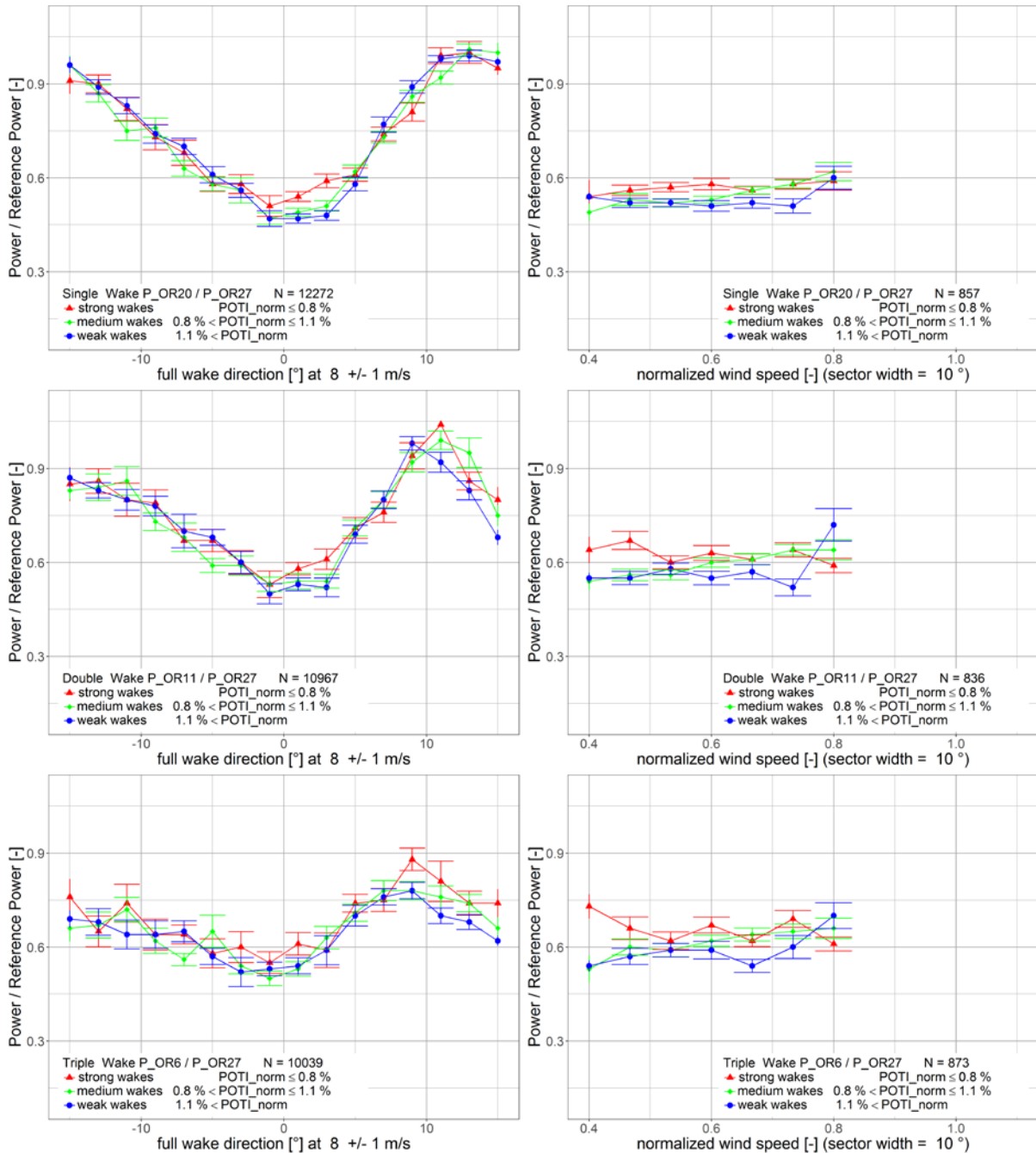

**Figure 19: Wake effects in Ormonde (OR) under different conditions. Power of downstream turbine normalised with free flow turbine. First row: single wake, second row: double wake and third row: triple wake. Left column: Normalised Power as function of wind direction, right column: Normalised power as function of wind speed.**

## 5 Conclusions

Measured data from three different offshore wind farms, two met masts, one buoy and one long range LiDAR has been analysed to identify different influence on power production at turbines operating in the wake. We have validated the method described in Dörenkämper (2015), which proposes to use the turbulence intensity, to describe the power production in the wake and compared it with the atmospheric stability evaluation proposed by (Ott and Nielsen, 2014). In this case, turbulence intensity could better distinguish between the magnitude of wake effects. A correlation analysis was performed and for wind speeds in partial load operation, the standard deviation of the power divided by its average power ($PO_{TI}$) was identified having similar behaviour than the turbulence intensity. A sensitivity check for $PO_{TI}$ revealed very detailed responsiveness to increases in turbulences due to neighbouring turbines and wind farms. Effects from wind farm neighbours are detectable even more than 38 rotor diameter away. A strong wind speed dependency of this signal can be eliminated by normalization with a third order polynomial fitted to the data. A classification of different turbine behaviour based on this adjusted $PO_{TI_{norm}}$ was analysed and compared to the classification with turbulence intensity TI.

Both signals can distinguish between stronger and weaker wake effects. The magnitude of influence of the $PO_{TI_{norm}}$ signals on wake effects is dependent on the level of inflow turbulence intensity. Higher inflow turbulence has already a higher wake mixing and therefore the wake added turbulence has a less pronounced contribution.

Using $PO_{TI_{norm}}$ to predict wakes more accurate is a promising approach, but further investigations are necessary to take controller properties into account to fill the wind speed range beyond the rated wind speed of the turbine in free flow conditions.

## Acknowledgements

The presented work is partly funded by the Commission of the European Communities, Research Directorate-General within the scope of the project "ClusterDesign" (Project No. 283145 (FP7 Energy)). We would like to thank Deutsche Offshore-Testfeld und Infrastruktur GmbH & Co. KG (DOTI), Research at alpha ventus (RAVE), Forschungsplattformen in Nord und Ostsee (FINO1), Innogy SE, Vattenfall Wind Power and Senvion SE for making this investigation possible. Furthermore a special thanks to the R Core Team for developing the open source language R (R_Core_Team, 2015).

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
