# Peer review of "An analysis of offshore wind farm SCADA measurements to identify key parameters influencing the magnitude of wake effects"

_Wind Energy Science, 2016_

## Referee Comment (RC1) · Anonymous Referee #1 · 10 Mar 2017

The manuscript entitled "An analysis of offshore wind farm SCADA measurements to identify key parameters influencing the magnitude of wake effects" deals with the using of operating information supplied by the wind turbines to assess the atmospheric stability conditions and then to make some conclusions about the wake interaction effects. The objective is fully relevant: wind farms, and particularly offshore ones, are not equipped with meteorological measurements to determine the real-time and reliable meteorological conditions (wind speed, wind direction and particularly atmospheric stability). On the other hand, wind farm models need field data to be validated. The authors attempt to find an indirect way to assess atmospheric stability in order to determine the magnitude of the expected wake effects, according to this parameter.

[Figure]

On the other hand, the methodology used in this manuscript to obtain the presented conclusions does not sound rigorous enough at this stage to be published in a journal. Some hypothesis are too strong and the methodology is not validated.

Please find below the arguments to justify the recommendation:

- A direct correlation is expected between the turbulence intensity and the atmospheric stability. Though, for a fixed stability condition, turbulence intensity can have big scatter and particularly at low wind speeds. Reference to the works from Dorenkampfer et al. (2012 and 2015) are used to justify this strong simplification but these references are a PhD thesis and a proceeding from a national conference. I would suggest to make references to publications in peer-review journals and to develop the arguments that give the possibility to reduce the stability effect to a turbulence intensity effect, and particularly at low Wind speeds.

- LiDAR measurements at Nord See wind farm NO : PPI planes are described as horizontal. LiDAR is located on the helicopter platform from the wind turbine NO48. One therefore guess that is corresponds to an altitude close to the hub height. Consequently, the laser beam should meet the wind turbine rotors NO44 and NO45, leading to unusable data in the vicinity of both rotors. On the other hand, on Figure 4, the visualizations of the velocity field, as well as the normalized velocity evolution versus the downwind distance do not present any unresolved areas close to the rotors. The velocity induction through the rotor is presented and discussed. Please explain how these data were reconstructed close to the rotors.

- §3.3 New classification and validation. This part is confusing. The authors determine a classification of the wake effect on the basis of the median of the normalized power of a wind turbine in wake interaction. It means that the intensity of the wake effect is determined by its statistical occurrence and not by its strength. Please elaborate an argumentation to justify this strategy of classification

- §4.2 Correlation analysis. It is not clear whether the data are sorted only according

to the turbulence intensity or also to the wind speed (as performed in Fig. 5). If the data are not sorted according to the wind speed, it means that different operating conditions are plotted without distinction in this correlation matrix. How can one expect to get strong correlations between data coming from the incoming flow conditions (fully independent of the operation parameters) and operation-driven data coming from the wind turbines without any additional filters ? Could you please show the evolution of the relative power fluctuation PO_TI with the WT power or with the wind speed? Regardless of this crucial point, one cannot state that the level of correlation is acceptable in order to use this information as a representation of the turbulence intensity, and even less of the atmospheric stability.

- §4.3.1 New classification and validation on Alpha Ventus. By applying the new classification; discrepancies in the power production due to the assessed stability is rather small and difficult to interpret. By comparing Fig 5 and Fig 9, one notice that the frequency of occurrence of each stability class is also totally dependent of the classification method. For instance, on the top-right plot, the unstable case occurrence is 13% of samples for the new classification, instead of 56% with the classification based on turbulence intensity. It show again the poor correlation between both information.

- The thresholds used in the new classification are different for each tested wind farms. It is justified by the fact that the wind turbines are different. But how can one explain that thresholds are different on the same wind farm (Ormonde) but for different wind directions? Please elaborate on this point.

---

## Referee Comment (RC2) · Anonymous Referee #2 · 28 Mar 2017

**General comments**

This paper presents a new parameterization of stability classes for the prediction of single and multiple wake effects based on met mast and LiDAR data. After reviewing the paper, I am fairly convinced that this line of reasoning is worth pursuing. However, there are some issues to be addressed before the paper can be recommended for publication. These are enumerated below.

**Specific comments**

**Page 2, line 12** 'rotordiameter' is missing a space.

**Page 3, fig 2 caption** 'cycles' should be 'circles'.

**Page 4, lines 11 and 12** There are two instances where 'is' should be 'are'.

**Page 4, lines 17 and 18** There are two spelling errors: 'allowes' and 'includs'. Also the possessive is not necessary for 'turbine' and 'nacelle'.

**Page 6, section 3.1** It is surprising that a study specifically considering stability effects is relying on a simplified classification technique. This introduces a considerable amount of unnecessary uncertainty as an independent variable (*i.e.* the stability) is not directly measured.

**Page 7, table 1** More discussion for the boundary values for TI is needed. How exactly were these values assigned to unstable, neutral, and stable?

**Page 7, eq 4** I suspect, but cannot verify, that the decent correlation between the two may be (in part) happenstance. Atmospheric turbulence intensity decreases with wind speed as a result of flow physics. The standard deviation of output power to power also decreases with wind speed but largely because the turbine controller plays an increasingly active role. The authors allude to this later in the paper but more discussion of why eq 4 might be a suitable proxy for eq 1 would be of interest.

**Page 7, line 26** Why use the median instead of the mean?

**Page 8, line 2** '...the thresholds are selected to achieve the best distinction between the three data sets.' As these thresholds are central to the stability classification (and this work in general), a mathematical definition of *best distinction* must be included. Currently, this work is unreproducible by a third party.

**Page 8, line 7**  Please avoid use of the word 'prove' in this context.

**Page 9, fig 5, bottom left**  One point just right of the centre for the stable curve is clearly an outlier. Any comment?

**Tables 2–5**  The large variation in thresholds suggests that the approach is not general and more details regarding how these thresholds are determined is needed.

**Page 16, line 19**  Why limit the data set to 7–9 m/s? Are these results representative?

**Page 17, line 1**  'The difference in power production between stable and unstable cases is in the range of 10%.' Stability is only inferred here from TI so the statement should preferably refer to differences between high and low TI conditions.

**Page 17, line 8**  The paper would be significantly more complete with this range of important wind speeds included in the analysis.

**Page 18, line 7**  Typo: 'reviled' should be 'revealed'.

---

## Author Comment (AC2) · 25 Apr 2017

Answers to comments from anonymous Referee #2 by

Niko Mittelmeier et al. April 25, 2017

Dear Referee,
Thank you very much for reviewing our paper. Your comments helped us to understand where certainly more explanations is needed and we hope that we could add clarity and additional content to answer your questions sufficiently. You are right, when you point out, that it's a big step from SCADA data to stability classification and that strong simplifications have been made. For this reason we have accessed more data to close the gap between meteorological stability classification, TI, Shear and SCADA signals. We also want to be more precise on the purpose of this work. The main aim is to find turbine signals which can describe the magnitude of wake effects that are varying with different environmental conditions. With these signals it should be possible to fine tune wake models for more accurate predictions.
Our responses to your comments are marked as ***/ Response /***.

This paper presents a new parameterization of stability classes for the prediction of single and multiple wake effects based on met mast and LiDAR data. After reviewing the paper, I am fairly convinced that this line of reasoning is worth pursuing. However, there are some issues to be addressed before the paper can be recommended for publication. These are enumerated below.

**Specific comments**

Page 2, line 12 'rotordiameter' is missing a space.

***/ changed to "rotor diameter" /***

Page 3, fig 2 caption 'cycles' should be 'circles'.

***/ changed to "circles" /***

Page 4, lines 11 and 12 There are two instances where 'is' should be 'are'.

***/both instances changed to "are" /***

Page 4, lines 17 and 18 There are two spelling errors: 'allowes' and 'includs'. Also the possessive is not necessary for 'turbine' and 'nacelle'.

***/ Changed to "allows" and "includes" and possessives deleted/***

Page 6, section 3.1 It is surprising that a study specifically considering stability effects is relying on a simplified classification technique. This introduces a considerable amount of unnecessary uncertainty as an independent variable (i.e. the stability) is not directly measured.

***/ You are right. An acceptable representation of stability is needed and therefore we have accessed new data that has been just recently published on the BSH Fino Server. We will also develop our arguments slightly different. A new reproducible classification (see your comment from page 8) based on the magnitude of wake effects will be used and predictability with the different measured signals ($\zeta = z/L$, turbulence intensity and Turbine SCADA) is studied.

With latest calibrated temperature data from DEWI (Richard Fruehmann) we were able to follow the approach suggested by Ott and Nielsen (2014) and calculated the dimensionless $\zeta = \frac{z}{L}$ for T_air at 33m. The plot below shows the data availability for the selected period.

[Figure]

We decided to keep the number of classes at three ("unstable", "neutral" and "stable") based on the following table: (We will add more description of the methodology for the estimation of thresholds in Section 3.3, see also your comment for page 8) The estimated thresholds have also been proposed by Rajewski et al. (2013).

| Category | Range |
|---|---|
| Unstable | $\zeta < -0.05$ |
| Neutral | $-0.05 \leq \zeta \leq 0.05$ |
| Stable | $\zeta > 0.05$ |

This leads to the following histogram for the three classes.

[Figure]

References:

Ott, S. and Nielsen, M.: Developments of the offshore wind turbine wake model Fuga, E-0046 Report 2014, DTU Wind Energy, Lyngby, Denmark., 2014.

Rajewski, D. A., Takle, E. S., Lundquist, J. K., Oncley, S., Prueger, J. H., Horst, T. W., Rhodes, M. E., Pfeiffer, R., Hatfield, J. L., Spoth, K. K., Doorenbos, R. K., Rajewski, D. A., Takle, E. S., Lundquist, J. K., Oncley, S., Prueger, J. H., Horst, T. W., Rhodes, M. E., Pfeiffer, R., Hatfield, J. L., Spoth, K. K. and Doorenbos, R. K.: Crop Wind Energy Experiment (CWEX): Observations of Surface-Layer, Boundary Layer, and Mesoscale Interactions with a Wind Farm, Bull. Am. Meteorol. Soc., 94(5), 655–672, doi:10.1175/BAMS-D-11-00240.1, 2013.

/***

Page 7, table 1 More discussion for the boundary values for TI is needed. How exactly were these values assigned to unstable, neutral, and stable?

***/ The presented values are taken from Dörenkämper (2015). There is no description how exactly these values have been selected. In earlier studies (Dörenkämper et al., 2012, 2014) the authors used 1min averaged data and therefore also different thresholds than in his 2015 studies.
With your later comment, requesting a more solid method to reproduce the assessment of the thresholds we will use our new classification method (See Comment for Page 8) .

References:

Dörenkämper, M., Tambke, J., Steinfeld, G., Heinemann, D. and Kühn, M.: Influence of marine boundary layer characteristics on power curves of multi megawatt offshore wind turbines, in Proceedings of 11th German Wind Energy Conference, Bremen, Germany, 7-8 November., 2012.

Dörenkämper, M., Tambke, J., Steinfeld, G., Heinemann, D. and Kühn, M.: Atmospheric Impacts on Power Curves of Multi-Megawatt Offshore Wind Turbines, J. Phys. Conf. Ser., 555(1), 12029, doi:10.1088/1742-6596/555/1/012029, 2014.

Dörenkämper, M.: An investigation of the atmospheric influence on spatial and temporal power fluctuations in offshore wind farms, PhD Thesis, University of Oldenburg, Oldenburg., 2015.

/***

Page 7, eq 4 I suspect, but cannot verify, that the decent correlation between the two may be (in part) happenstance. Atmospheric turbulence intensity decreases with wind speed as a

result of flow physics. The standard deviation of output power to power also decreases with wind speed but largely because the turbine controller plays an increasingly active role. The authors allude to this later in the paper but more discussion of why eq 4 might be a suitable proxy for eq 1 would be of interest.

***/ This is a very valid point. Therefore we have investigated more on the influence of wind speed. In the plot below, we have used the new stability classification based on $\zeta = z/L$ (See comment for Page 6). Bin averaged turbulence intensity (TI_100) measured at the met mast and at the nacelle (AV4_TI)  as well as met mast shear (alpha_90_40)  and AV4_POTI are plotted for each class as function of the wind speed. The selected bin of $8 \pm 1$m/s is quite well distinguishable with constant thresholds for all the provided variables. Whereas turbulence intensity from the met mast is fairly constant for the selected wind speed range,  shear and POTI are showing a strong dependency on wind speed.

[Figure]

For this reason we propose to develop Eq. 5 which takes wind speed measured at the nacelle also into account.

We are now proposing a correction for POTI to substitute the wind speed dependency. This can be done by normalizing POTI with a third order polynomial. The resulting plot is shown below.

[Figure]

/***

Page 7, line 26 Why use the median instead of the mean?

***/ In our data example we obtained a mean = 0.516 and a median = 0.5108 which is very close together (0.0052). We decided to use the median because the mean was effected by some outliers. A deeper analysis of these outliers revealed that an additional filter criteria for the data is needed. The new filter removes 10-min intervals when one of the turbines has had a downtime in the interval before. In this way the flow throw the wind farm gets another 10-min time to develop. Additionally data with a power ratio > 1 meaning that the turbine in the wake center (±5°) produces more than a free flow turbine has been deleted (only two values). After removing these outliers, mean and median have now a difference of 0.0015. We agree that it is more appropriate to use the mean when enough care for outliers has been taken.

We will describe the new filtering in 2.4 and change 3.3 to "mean" instead of median.
/***

Page 8, line 2 '...the thresholds are selected to achieve the best distinction between the three data sets.' As these thresholds are central to the stability classification (and this work in general), a mathematical definition of best distinction must be included. Currently, this work is unreproducible by a third party.

***/ This is a very valid point. We will describe the methodology in more detail:

At first we select the normalized power (waked turbine, normalized by the power of a free flow turbine) for a small sector (10°) in the full wake for the relevant wind speed range (8 ± 1 m/s) (Fig 1a). Secondly we eliminate the dependency on wind direction by normalizing the normalized power for each wind direction bin (binwidth = 2°) with its mean value (Fig. 1b).

[Figure]

Fig. 1a

Fig. 1b

The third step divides the data set into high wake effects (values < 0) and low wake effects (values >= 0) and the density distribution of the variable of interest is plotted for these two data sets (Fig 2). We use the median for each density distribution to allocate the thresholds.

[Figure]

Fig 2. Data density for different variables based on low and high wake effects. The median for each distribution is highlighted with a vertical line. The data corresponds to a wind speed bin of 8 ± 1 m/s and a sector width of 10° around the full wake.

Note:

The TI and POTI thresholds have slightly changed compared to our first version of the paper. The difference in TI thresholds is due to the fact, that we have used the values from Dörenkämper (2015) and now we are suggesting this new methodology.

POTI thresholds have also slightly changed because the criteria was visual inspected and now we propose to use the median. In this way, the results should be reproducible now.

To overcome the shortfall of AV4_POTI signal having a strong dependency on wind speed, we propose a normalization of this signal with a third order polynomial.

[Figure]

When applying the same methodology to AV4_POTI_norm as described above, we obtain a density distribution as below:

[Figure]

The table below summarizes different classes of interest:

| Category | $\zeta = z/L$ [-] | TI_100 [%] | AV4_POTI_norm [-] |
|---|---|---|---|
| Weak wakes | $\zeta < -0.05$ | TI_100 $< 5.4\%$ | $POTI_{norm} < 0.8$ |
| Medium wakes | $-0.05 \leq \zeta \leq 0.05$ | $5.4 \leq T_{100} \leq 6.5$ | $0.8 \leq POTI_{norm} \leq 1.1$ |
| Strong wakes | $\zeta > 0.05$ | $T_{100} > 6.5$ | $POTI_{norm} > 1.1$ |

Looking at the distributions for each class, one can see an improvement from POTI to POTI_norm. Latter is much more comparable to the turbulence intensity measured at the met mast.

For z/L, weak wakes cases seem to become less frequent with increasing wind speed. POTI seams to overestimate this trend. TI_100 and AV4_POTI_norm provide similar results.

[Figure]

Using AV_POTI_norm as a classifier, we obtain the following wake plot:

[Figure]

References:

Dörenkämper, M.: An investigation of the atmospheric influence on spatial and temporal power fluctuations in offshore wind farms, PhD Thesis, University of Oldenburg, Oldenburg., 2015.
/***

Page 8, line 7 Please avoid use of the word 'prove' in this context.

***/ "prove" replaced by "shows"/***

Page 9, fig 5, bottom left One point just right of the centre for the stable curve is clearly an outlier. Any comment? Tables 2–5 The large variation in thresholds suggests that the approach is not general and more details regarding how these thresholds are determined is needed.

***/ We have inspected the outlier and added two new filter criteria. The new filter removes 10-min intervals when one of the turbines has had a downtime in the interval before. In this way the flow throw the wind farm gets another 10-min time to develop. Additionally data with a power ratio > 1 meaning that the turbine in the full wake (wake center ±5°) produces more than a free flow turbine has been removed.

The new plot:

[Figure]

/***

Page 16, line 19 Why limit the data set to 7–9 m/s? Are these results representative?

***/We will change the wording from "7-9 m/s" to "8 ± 1 m/s". In this way it is consistent to the Fig. 5, Fig. 9 and Fig. 13. /***

Page 17, line 1 'The difference in power production between stable and unstable cases is in the range of 10%.' Stability is only inferred here from TI so the statement should preferably refer to differences between high and low TI conditions.

***/Agreed!
New wording:
"The difference in power production between high and low turbulence conditions is in the range of 10%, …/***

Page 17, line 8 The paper would be significantly more complete with this range of important wind speeds included in the analysis.

***/ That's true, we are working on it/***

Page 18, line 7 Typo: 'reviled' should be 'revealed'.

***/changed to "revealed"/***

---

## Author Response (AR1)

[revised manuscript text omitted]

Kommentar [RC2-2]: Page 2, line 12 'rotordiameter' is missing a space.

[Figure]

**Figure 1:** Scematic Layout of the Nnorthern part of alpha ventus wind farm (circles) and FINO1 met mast (red square) layout with free flow sector and distance in rotor diameters

**2.2 Nordsee Ost**

The wind farm Nordsee Ost (NO) is located about 35 km north-west of the island of Helgoland in the North Sea. The 48 Senvion turbines have a rated power of 6 MW each and a rotor diameter of 126 m. The met mast is located in the south-western corner of the wind farm (See Fig. 2). In the south, the neighbouring wind farm Meerwind Ost/Süd reduces the sector of free flow for the met mast as well as the possibilities to study multiple wakes higher than triple wake condition without disturbing effects from Meerwind.

The wind farm Nordsee Ost has been fully commissioned in 2015. So far not enough dataData for this analysis is selected from (11/2015 – 11/2016.) has been collected to investigate the full wake behaviour based on SCADA data. For this reason, a A correlation analysis (described in SectionSect. 3.2) is performed and the data from the a ClusterDesign long range LiDAR measurement campaign is analysed. This LiDAR measurement campaign took place within the European Research Project "ClusterDesign".

[Figure]

**Figure 2: Nordsee Ost  (blue circles) with neighbouring wind farm Meerwind Süd (green triangles)  ,met mast (red square) and distance in rotor diameter (D). Orange area indicates the Plan Position Indicator (PPI) scan from the Windcube 200S, mounted on the helicopter platform of NO48. (Described in Sect. 2.5)**

Kommentar [RC2-3]: Page 3, fig 2 caption 'cycles' should be 'circles'.

**2.3 Ormonde**

The Ormonde wind farm consists of 30 Senvion turbines with a rated power of 5 MW each and a rotor diameter of 126 m. The wind farm is located in the Irish Sea 10 km west of the Isle of Walney. The selected data is from 1/2012 – 1/2014. During this period, neighbouring wind farms were operational. Located In the south west are Walney1 (51 x SWT-3.6-107 Siemens) and Walney 2 (51 x SWT-3.6-120 Siemens), located in the south there is West of Duddon Sands (SWT-3.6-120 Siemens, fully commissioned 30.10.2014) and in the south east there is Barrow (V90 3.0MW Vestas).

The farm layout displayed in Fig. 3 is structured in a regular array which allows for comparing several multiple-wake situations. The inner farm turbine distance for the investigated wake situation from south west is 6.3 D and from north west is 4.3 D.

[Figure]

**Figure 3: a) Ormonde and neighbouring wind farms .  b) Ormonde wind farm layout with distances in rotor diameters (D) and sectors selected for the analysis.**

**2.4 SCADA and meteorological data**

The SCADA data from all wind farms and the meteorological data consist of 10-min statistics. Each turbine provides wind speed, wind direction, active power, yaw position, and pitch angle. The operational condition of the wind turbine which is used for the correlation with the met mast turbulence intensity is categorized by the minimum active power > 10kW, the maximum pitch angle < 3 ° and the standard deviation of the yaw position < 5 °. These filter criteria's ensure that no stand stills, curtailments or too large yaw activities are included in the data. Furthermore any succeeding 10-min measurement period after a turbine restart is deleted to give the flow enough time to develop. SCADA data are also deleted if the waked turbine produces more power than the free flow turbine. Implausible met mast data  are removed and wind directions  are corrected for bias by using the orientation of the maximum wake deficit. For the correlation, only sectors of free flow conditions  are used. Averages of 30-min water temperature are recorded by buoys at FINO1 and linearly interpolated into the SCADA data.

**2.5 Long range LiDAR measurements**

Within the "ClusterDesign" research project, funded by the European Union, a long-range LiDAR measurement campaign was realized. A Windcube 200S (WLS200S) LiDAR with scan head was placed on the helicopter platform of NO48 (See Fig. 2) from 11/2015 – 5/2016. A differential GPS system composed by three antenna GNSS-Systems of type Trimble SPS855 / SPS555H allows for additional measurements of turbines yaw and nacelle pitch and roll angle. One LiDAR measurement cycle takes about 200-s. It includes five  Plan  Position  Indicator (PPI) scans followed by one  Range  Height  Indicator (RHI) scan. Both scans cover a sector of 30-° on the

horizontal and vertical plane respectively and are centred on the rotor axis. The scan trajectories have an angular resolution of 1 ° and measured the wind speed component along the measuring direction every 25 m from 100 m to 2500 m. The LiDAR data is filtered excluding measurements with a poor signal intensity, or affected by hard targets , orand considered outliers.

[Figure]

**Kommentar [RC1-6]:** - LiDAR measurements at Nord See wind farm NO : PPI planes are described as horizontal. LiDAR is located on the helicopter platform from the wind turbine NO48. One therefore guess that is corresponds to an altitude close to the hub height. Consequently, the laser beam should meet the wind turbine rotors NO44 and NO45, leading to unusable data in the vicinity of both rotors. On the other hand, on Figure 4, the visualizations of the velocity field, as well as the normalized velocity evolution versus the downwind distance do not present any unresolved areas close to the rotors. The velocity induction through the rotor is presented and discussed. Please explain how these data were reconstructed close to the rotors.

[revised manuscript text omitted]

**Kommentar [RC2-14]:** Page 9, fig 5, bottom left One point just right of the centre for the stable curve is clearly an outlier. Any comment? Tables 2–5 The large variation in thresholds suggests that the approach is not general and more details regarding how these thresholds are determined is needed.

[Figure]

**Figure 9: Wake effects in alpha ventus (AV) under different atmospheric conditions classified by $\zeta = 33/L$. Power of downstream turbine normalised with free flow turbine. Upper row: single wake, bottom row: double wake. Left column: Normalised Power as function of wind direction, right column: Normalised power as function of wind speed.**

**Kommentar [RC2-15]:** Page 9, fig 5, bottom left One point just right of the centre for the stable curve is clearly an outlier. Any comment? Tables 2–5 The large variation in thresholds suggests that the approach is not general and more details regarding how these thresholds are determined is needed.

[Figure]

**Figure 10: Wind speed recovery at wake centre on hub height behind NO48 for different turbulence stability classes. The wind speed is normalized with the inflow wind speed and the distance from the LiDAR on NO48 downstream is displayed in multiples of rotor diameters**

**4.2 Correlation analysis**

In the next step, we correlate the SCADA signals described in Sect. 3.2 with the turbulence intensity measured at the mast. In Fig. 7 a panel plot is displayed. The graphs on the diagonal present the histogram and density distribution for the respective variable. The panels above the diagonal provide the Pearson correlation coefficients. The lower panels are scatter plots for the two variables with a fitted linear regression line. The colours of the points indicate the three stability classifications (blue: stable, green: neutral, red: unstable) determined with the met mast turbulence intensity.

The correlation between met mast and turbine TI in subplot (1 , 2) equals to 0.55. This poor result can be explained by the nacelle wind speed measurement position behind the rotor, which induces additional disturbance to the flow.

The highest correlation with the met mast TI is obtained with the standard deviation of the turbine power divided by its average active power ($PO_{TI,AV4}$) in subplot (1 , 5). Although a correlation of 0.66 is not perfect, it is still better than the turbulence measured with the nacelle cup anemometer. Especially in the low turbulence region, the scatter plot proves to be denser. Very similar results are obtained when applying the same analysis to AV1 and AV2.

**Kommentar [RC1-16]:** - §4.2 Correlation analysis. It is not clear whether the data are sorted only according to the turbulence intensity or also to the wind speed (as performed in Fig. 5). If the data are not sorted according to the wind speed, it means that different operating conditions are plotted without distinction in this correlation matrix. How can one expect to get strong correlations between data coming from the incoming flow conditions (fully independent of the operation parameters) and operation-driven data coming from the wind turbines without any additional filters ? Could you please show the evolution of the relative power fluctuation PO_TI with the WT power or with the wind speed? Regardless of this crucial point, one cannot state that the level of correlation is acceptable in order to use this information as a representation of the turbulence intensity, and even less of the atmospheric stability.

**Kommentar [NM17]:** Please find the requested plot in Fig.7. The filtering is described in Sect. 2.4.

[revised manuscript text omitted]

**Kommentar [RC1-18]:** - §4.3.1 New classification and validation on Alpha Ventus. By applying the new classification; discrepancies in the power production due to the assessed stability is rather small and difficult to interpret. By comparing Fig 5 and Fig 9, one notice that the frequency of occurrence of each stability class is also totally dependent of the classi- fication method. For instance, on the top-right plot, the unstable case occurrence is 13% of samples for the new classification, instead of 56% with the classification based on turbulence intensity. It show again the poor correlation between both information.

[revised manuscript text omitted]

Wind Energ. Sci. Discuss., doi:10.5194/wes-2016-60-RC1, 2017 © Author(s) 2017. CC-BY 3.0 License.
Thank you very much for reviewing our paper. Your concern, that our results are only usable for the selected wind speed bin is valid. The proposed signal is dependent on wind speed and therefore a constant threshold is limited useful. We will provide an adjustment to this fact in our answers to your comments. We will also follow your suggestion to develop the arguments in a new way. We focus on the classification of the magnitude of wake effects and show the ability to predict these conditions with the different signals. Your comments helped us to understand where we need to provide more clarity and we hope that our answers to your questions will improve the paper. Our responses to your comments are marked as ***/ Response /***.

The manuscript entitled "An analysis of offshore wind farm SCADA measurements to identify key parameters influencing the magnitude of wake effects" deals with the using of operating information supplied by the wind turbines to assess the atmospheric stability conditions and then to make some conclusions about the wake interaction effects. The objective is fully relevant: wind farms, and particularly offshore ones, are not equipped with meteorological measurements to determine the real-time and reliable meteorological conditions (wind speed, wind direction and particularly atmospheric stability). On the other hand, wind farm models need field data to be validated. The authors attempt to find an indirect way to assess atmospheric stability in order to determine the magnitude of the expected wake effects, according to this parameter.

On the other hand, the methodology used in this manuscript to obtain the presented conclusions does not sound rigorous enough at this stage to be published in a journal. Some hypothesis are too strong and the methodology is not validated.

Please find below the arguments to justify the recommendation:

- A direct correlation is expected between the turbulence intensity and the atmospheric stability. Though, for a fixed stability condition, turbulence intensity can have big scatter and particularly at low wind speeds. Reference to the works from Dörenkämper et al. (2012 and 2015) are used to justify this strong simplification but these references are a PhD thesis and a proceeding from a national conference. I would suggest to make references to publications in peer-review journals and to develop the arguments that give the possibility to reduce the stability effect to a turbulence intensity effect, and particularly at low Wind speeds.

***/ You are right, using only turbulence intensity (TI) is a very strong simplification for stability. To come up to the readers expectations we will add more clarity to the abstract and introduction to make the purpose of the paper more defined.

The idea behind this research is to find a signal that could be used to improve wake model tuning for specific operational conditions. This may help to improve the use of a wake model to detect underperformance as proposed in Mittelmeier et al. (2017). We will show, that the magnitude of wake effect is not only governed by stability but also turbulence intensity and we will show to which extend one can expect an overlapping effect.

For this purpose, we will provide peer reviewed journal publications to base the assumptions on a more solid foundation and evaluate new data to be able to compare a stability description defined by the Monin-Obukhov similarity theory and the SCADA data.

In Hansen et al. (2012), the authors studied wake effects at Horns Rev in different atmospheric conditions. They also compared turbulence intensities for different stability classes as a function of the wind speed. Below 7m/s a clear increase in TI can be noticed. Above 7m/s neutral-unstable conditions are clearly distinguishable from more stable conditions with a constant threshold up to nominal wind speed. Dörenkämper et al. (2014) published their results also at a peer reviewed conference series where they draw the link from stability via shear to turbulence intensity motivated by the studies of (Tambke et al., 2005). In a later study, Sanz Rodrigo et al. (2015) compared different stability classification methodologies with data from FINO 1 and presented the behavior of shear and turbulence intensity for the proposed atmospheric stability classes. The authors concluded, that in this particular cases TI correlates well for stable cases but at near neutral and unstable cases, shear is supposed to enable better distinction between their nine classes.

We investigated new data from FINO1 to be able to use a "real" stability classification and not only TI classes.

With latest calibrated temperature data from DEWI (Richard Fruehmann) we were able to follow the approach suggested by Ott and Nielsen (2014) and calculated the dimensionless $\zeta = \frac{z}{L}$ for T_air at 33m. The plot below shows the data availability for the selected period.

[Figure]

We decided to keep the number of classes at three ("unstable", "neutral" and "stable") based on the following table:

| Category | Range |
|---|---|
| Unstable | $\zeta < -0.05$ |
| Neutral | $-0.05 \leq \zeta \leq 0.05$ |
| Stable | $\zeta > 0.05$ |

(We will add more description of the methodology for the estimation of thresholds in Section 3.3) The estimated thresholds have also been proposed by Rajewski et al. (2013).
This leads to the following histogram for the three classes.

[Figure]

In the plot below, we have used the new stability classification based on $\zeta = z/L$. Bin averaged turbulence intensity (TI_100) measured at the met mast and at the nacelle (AV4_TI) as well as met mast shear (alpha_90_40) and AV4_POTI are plotted for each class as function of the wind speed. The selected wind speed bin of $8 \pm 1$m/s is quite well distinguishable with constant thresholds for all the provided variables. But whereas turbulence intensity from the met mast is fairly constant for the whole presented range, shear and POTI are showing a much stronger dependency on wind speed.

[Figure]

The plots above confirm the statement made by Tambke et al. (2005) and Sanz Rodrigo et al. (2015), that shear enables a more clear distinction between the atmospheric stability classes. The error bars, indicating one standard deviation are not overlapping anymore for stable and unstable class above 8 m/s.

As you have suggested, we will develop the arguments in a new way. We describe how to classify low and high wake effects and use this characteristic to evaluate the predictability based on stability, turbulence intensity and turbine SCADA data classes. Then we provide an overview on the different occurrences of the described environmental conditions.

Determination of thresholds:
At first we select the normalized power (waked turbine, normalized by the power of a free flow turbine) for a small sector (10°) in the full wake for the relevant wind speed range (8 ± 1 m/s) (Fig 1a). Secondly we eliminate the dependency on wind direction by normalizing the normalized power for each wind direction bin (binwidth = 2°) with its mean value (Fig. 1b).

[Figure]

Fig. 1a

Fig. 1b

The third step divides the data set into high wake effects (values < 0) and low wake effects (values >= 0) and the density distribution of the variable of interest is plotted for these two data sets (Fig 2). We use the median for each density distribution to allocate thresholds.

[Figure]

Fig 2. Data density for different variables based on low and high wake effects. The median for each distribution is highlighted with a vertical line. The data corresponds to a wind speed bin of 8 ± 1 m/s and a sector width of 10° around the full wake.

Note:
The TI and POTI thresholds have slightly changed compared to our first version of the paper. The difference in TI thresholds is due to the fact, that we have used the values from Dörenkämper (2015) and now we are suggesting this new methodology.
POTI thresholds have also slightly changed from the old version of our paper because the criteria was visual inspected and now we propose to use the median. In this way, the results should be reproducible now.

To overcome the shortfall of AV4_POTI signal having a strong dependency on wind speed, we propose a normalization of this signal with a third order polynomial.

[Figure]

When applying the same methodology to AV4_POTI_norm as described above, we obtain density distributions as below:

[Figure]

The table below summarizes the different classes:

| Category | $\zeta = z/L$ [-] | TI_100 [%] | AV4_POTI_norm [-] |
|---|---|---|---|
| Weak wakes | $\zeta < -0.05$ | TI_100 $< 5.4\%$ | $POTI_{norm} < 0.8$ |
| Medium wakes | $-0.05 \leq \zeta \leq 0.05$ | $5.4 \leq T_{100} \leq 6.5$ | $0.8 \leq POTI_{norm} \leq 1.1$ |
| Strong wakes | $\zeta > 0.05$ | $T_{100} > 6.5$ | $POTI_{norm} > 1.1$ |

Looking at the distributions for each class, one can see an improvement from POTI to POTI_norm. Latter is much more comparable to the turbulence intensity measured at the met mast.

[Figure]

For z/L, weak wakes cases seem to become less frequent with increasing wind speed. POTI seams to overestimate this trend. TI_100 and AV4_POTI_norm provide similar results. Using AV_POTI_norm as a classifier, we obtain the following wake plot:

[Figure]

/***

- LiDAR measurements at Nord See wind farm NO : PPI planes are described as horizontal. LiDAR is located on the helicopter platform from the wind turbine NO48. One therefore guess that is corresponds to an altitude close to the hub height. Consequently, the laser beam should meet the wind turbine rotors NO44 and NO45, leading to unusable data in the vicinity of both rotors. On the other hand, on Figure 4, the visualizations of the velocity field, as well as the normalized velocity evolution versus the downwind distance do not present any unresolved areas close to the rotors. The velocity induction through the rotor is presented and discussed. Please explain how these data were reconstructed close to the rotors.

***/ Yes, you are right. Hard targets like blades and nacelle prevent a reasonable wind speed measurement. Due to the fact, that we are using multiple PPI-scans to obtain a 10-min average value, there are wind speed measurements available in the rotor plane.
To derive the wind speeds for Fig. 4 (top), a raster layer was generated. While raster cells with multiple measurements are averaged, values with empty cells are linearly interpolated. In this way, the blind region at each nacelle and areas between the beam directions show interpolated wind speeds.

We will add the following text:
"Hard targets like blades and nacelle prevent reasonable wind speed measurement. Wind speeds at these blind regions and between the beam directions are linearly interpolated. "
/***

- §3.3 New classification and validation. This part is confusing. The authors determine a classification of the wake effect on the basis of the median of the normalized power of a wind turbine in wake interaction. It means that the intensity of the wake effect is determined by its statistical occurrence and not by its strength. Please elaborate an argumentation to justify this strategy of classification

***/ You are right, this part has been not sufficiently described. In our data example we obtained a mean = 0.516 and a median = 0.5108 which is very close together (0.0052). We decided to use the median because the mean was effected by some outliers. A deeper analysis of these outliers revealed that an additional filter criteria for the data is needed. The new filter removes 10-min intervals when one of the turbines has had a downtime in the interval before. In this way the flow throw the wind farm gets another 10-min time to develop. Additionally data with a power ratio > 1 meaning that the turbine in the wake produces more than a free flow turbine has been deleted (only two values). After removing these outliers, mean and median have now a difference of 0.0015. We agree that it is more appropriate to use the mean when enough care for outliers has been taken.

We will describe the new filtering in 2.4 and change 3.3 to "mean" instead of median.
/***

- §4.2 Correlation analysis. It is not clear whether the data are sorted only according to the turbulence intensity or also to the wind speed (as performed in Fig. 5). If the data are not sorted according to the wind speed, it means that different operating conditions are plotted without distinction in this correlation matrix. How can one expect to get strong correlations between data coming from the incoming flow conditions (fully independent of the operation parameters) and operation-driven data coming from the wind turbines without any additional filters ? Could you please show the evolution of the relative power fluctuation PO_TI with the WT power or with the wind speed? Regardless of this crucial point, one cannot state that the level of correlation is acceptable in order to use this information as a representation of the turbulence intensity, and even less of the atmospheric stability.

***/ You are right. Please find below the requested plots. Within the wind speed range of interest, the turbine signal is much more dependent on wind speed than the measured turbulence intensity at the met mast. All our wake plots are based on the wind speed bin 8 ± 1 m/s. In this range the correlation is higher, than for the other wind speeds.

[Figure]

/***

- §4.3.1 New classification and validation on Alpha Ventus. By applying the new classification; discrepancies in the power production due to the assessed stability is rather small and difficult to interpret. By comparing Fig 5 and Fig 9, one notice that the frequency of occurrence of each stability class is also totally dependent of the classification method. For instance, on the top-right plot, the unstable case occurrence is 13% of samples for the new classification, instead of 56% with the classification based on turbulence intensity. It show again the poor correlation between both information.

***/ In Fig. 9 the % values are biased because the "stable" class also contains all data where the turbine controller has already started with the main pitch activity which leads to a lower standard deviation of the power. We have to filter this part of the operation before displaying the proportions of the classes. We will call this part of the data "unclassified". The corrected plot is provided below:

[Figure]

We think this plot is not relevant anymore. We would like to show the new classification based on the normalized AV4_POTI_norm. (As provided for the answer of your first comment)
/***

- The thresholds used in the new classification are different for each tested wind farms. It is justified by the fact that the wind turbines are different. But how can one explain that thresholds are different on the same wind farm (Ormonde) but for different wind directions?

***/We assume that this has been an error due to the fact that POTI was very much dependent on wind speed. With the new proposed method to use a normalized values we can stick to the same classification even for different turbines, cause the turbine behavior is canceled out. /***

Wind Energ. Sci. Discuss., doi:10.5194/wes-2016-60-RC2, 2017 © Author(s) 2017. CC-BY 3.0 License.
Thank you very much for reviewing our paper. Your comments helped us to understand where certainly more explanations is needed and we hope that we could add clarity and additional content to answer your questions sufficiently. You are right, when you point out, that it's a big step from SCADA data to stability classification and that strong simplifications have been made. For this reason we have accessed more data to close the gap between meteorological stability classification, TI, Shear and SCADA signals. We also want to be more precise on the purpose of this work. The main aim is to find turbine signals which can describe the magnitude of wake effects that are varying with different environmental conditions. With these signals it should be possible to fine tune wake models for more accurate predictions.
Our responses to your comments are marked as ***/ Response /***.

This paper presents a new parameterization of stability classes for the prediction of single and multiple wake effects based on met mast and LiDAR data. After reviewing the paper, I am fairly convinced that this line of reasoning is worth pursuing. However, there are some issues to be addressed before the paper can be recommended for publication. These are enumerated below.

**Specific comments**

Page 2, line 12 'rotordiameter' is missing a space.

***/ changed to "rotor diameter" /***

Page 3, fig 2 caption 'cycles' should be 'circles'.

***/ changed to "circles" /***

Page 4, lines 11 and 12 There are two instances where 'is' should be 'are'.

***/both instances changed to "are" /***

Page 4, lines 17 and 18 There are two spelling errors: 'allowes' and 'includs'. Also the possessive is not necessary for 'turbine' and 'nacelle'.

***/ Changed to "allows" and "includes" and possessives deleted/***

Page 6, section 3.1 It is surprising that a study specifically considering stability effects is relying on a simplified classification technique. This introduces a considerable amount of unnecessary uncertainty as an independent variable (i.e. the stability) is not directly measured.

***/ You are right. An acceptable representation of stability is needed and therefore we have accessed new data that has been just recently published on the BSH Fino Server. We will also develop our arguments slightly different. A new reproducible classification (see your comment from page 8) based on the magnitude of wake effects will be used and predictability with the different measured signals ($\zeta = z/L$, turbulence intensity and Turbine SCADA) is studied.

With latest calibrated temperature data from DEWI (Richard Fruehmann) we were able to follow the approach suggested by Ott and Nielsen (2014) and calculated the dimensionless $\zeta = \frac{z}{L}$ for T_air at 33m. The plot below shows the data availability for the selected period.

[Figure]

We decided to keep the number of classes at three ("unstable", "neutral" and "stable") based on the following table: (We will add more description of the methodology for the estimation of thresholds in Section 3.3, see also your comment for page 8) The estimated thresholds have also been proposed by Rajewski et al. (2013).

| Category | Range |
|---|---|
| Unstable | $\zeta < -0.05$ |
| Neutral | $-0.05 \leq \zeta \leq 0.05$ |
| Stable | $\zeta > 0.05$ |

This leads to the following histogram for the three classes.

[Figure]

/***

Page 7, eq 4 I suspect, but cannot verify, that the decent correlation between the two may be (in part) happenstance. Atmospheric turbulence intensity decreases with wind speed as a

result of flow physics. The standard deviation of output power to power also decreases with wind speed but largely because the turbine controller plays an increasingly active role. The authors allude to this later in the paper but more discussion of why eq 4 might be a suitable proxy for eq 1 would be of interest.

***/ This is a very valid point. Therefore we have investigated more on the influence of wind speed. In the plot below, we have used the new stability classification based on $\zeta = z/L$ (See comment for Page 6). Bin averaged turbulence intensity (TI_100) measured at the met mast and at the nacelle (AV4_TI) as well as met mast shear (alpha_90_40) and AV4_POTI are plotted for each class as function of the wind speed. The selected bin of $8 \pm 1$m/s is quite well distinguishable with constant thresholds for all the provided variables. Whereas turbulence intensity from the met mast is fairly constant for the selected wind speed range, shear and POTI are showing a strong dependency on wind speed.

[Figure]

For this reason we propose to develop Eq. 5 which takes wind speed measured at the nacelle also into account.

We are now proposing a correction for POTI to substitute the wind speed dependency. This can be done by normalizing POTI with a third order polynomial. The resulting plot is shown below.

[Figure]

/***

Page 7, line 26 Why use the median instead of the mean?

***/ In our data example we obtained a mean = 0.516 and a median = 0.5108 which is very close together (0.0052). We decided to use the median because the mean was effected by some outliers. A deeper analysis of these outliers revealed that an additional filter criteria for the data is needed. The new filter removes 10-min intervals when one of the turbines has had a downtime in the interval before. In this way the flow throw the wind farm gets another 10-min time to develop. Additionally data with a power ratio > 1 meaning that the turbine in the wake center (±5°) produces more than a free flow turbine has been deleted (only two values). After removing these outliers, mean and median have now a difference of 0.0015. We agree that it is more appropriate to use the mean when enough care for outliers has been taken.

We will describe the new filtering in 2.4 and change 3.3 to "mean" instead of median.
/***

Page 8, line 2 '...the thresholds are selected to achieve the best distinction between the three data sets.' As these thresholds are central to the stability classification (and this work in general), a mathematical definition of best distinction must be included. Currently, this work is unreproducible by a third party.

***/ This is a very valid point. We will describe the methodology in more detail:

At first we select the normalized power (waked turbine, normalized by the power of a free flow turbine) for a small sector (10°) in the full wake for the relevant wind speed range (8 ± 1 m/s) (Fig 1a). Secondly we eliminate the dependency on wind direction by normalizing the normalized power for each wind direction bin (binwidth = 2°) with its mean value (Fig. 1b).

[Figure]

Fig. 1a

Fig. 1b

The third step divides the data set into high wake effects (values < 0) and low wake effects (values >= 0) and the density distribution of the variable of interest is plotted for these two data sets (Fig 2). We use the median for each density distribution to allocate the thresholds.

[Figure]

Fig 2. Data density for different variables based on low and high wake effects. The median for each distribution is highlighted with a vertical line. The data corresponds to a wind speed bin of $8 \pm 1$ m/s and a sector width of $10°$ around the full wake.

Note:

The TI and POTI thresholds have slightly changed compared to our first version of the paper. The difference in TI thresholds is due to the fact, that we have used the values from Dörenkämper (2015) and now we are suggesting this new methodology.

POTI thresholds have also slightly changed because the criteria was visual inspected and now we propose to use the median. In this way, the results should be reproducible now.

To overcome the shortfall of AV4_POTI signal having a strong dependency on wind speed, we propose a normalization of this signal with a third order polynomial.

[Figure]

When applying the same methodology to AV4_POTI_norm as described above, we obtain a density distribution as below:

[Figure]

The table below summarizes different classes of interest:

| Category | $\zeta = z/L$ [-] | TI_100 [%] | AV4_POTI_norm [-] |
|---|---|---|---|
| Weak wakes | $\zeta < -0.05$ | TI_100 $< 5.4\%$ | $POTI_{norm} < 0.8$ |
| Medium wakes | $-0.05 \leq \zeta \leq 0.05$ | $5.4 \leq T_{100} \leq 6.5$ | $0.8 \leq POTI_{norm} \leq 1.1$ |
| Strong wakes | $\zeta > 0.05$ | $T_{100} > 6.5$ | $POTI_{norm} > 1.1$ |

Looking at the distributions for each class, one can see an improvement from POTI to POTI_norm. Latter is much more comparable to the turbulence intensity measured at the met mast.
For z/L, weak wakes cases seem to become less frequent with increasing wind speed. POTI seams to overestimate this trend. TI_100 and AV4_POTI_norm provide similar results.

[Figure]

Using AV_POTI_norm as a classifier, we obtain the following wake plot:

[Figure]

Page 17, line 1 'The difference in power production between stable and unstable cases is in the range of 10%.' Stability is only inferred here from TI so the statement should preferably refer to differences between high and low TI conditions.

***/Agreed!
New wording:
"The difference in power production between high and low turbulence conditions is in the range of 10%, …/***

Page 17, line 8 The paper would be significantly more complete with this range of important wind speeds included in the analysis.

***/ That's true, we are working on it/***

Page 18, line 7 Typo: 'reviled' should be 'revealed'.

***/changed to "revealed"/***

---

## Author Response (AR2)

Answers to comments of anonymous Referee #1 and Editor-in-chief, Jakob Mann by

Niko Mittelmeier et al. September 08, 2017

Dear Referee, dear Jakob,
Thank you very much for reviewing our paper. We have made all proposed language corrections.

"dependency" -> "dependence" Changed everywhere
l3: "at turbines" -> "for turbines" Changed
l4: "intensity, to" -> "intensity to" Changed
l5: "by (Ott and Nielsen, 2014)" -> "by Ott and Nielsen (2014)" Changed
l8: "behaviour than" -> "behaviour to that of" Changed
l9: "in turbulences" -> "in turbulence" Changed
l10: "rotor diameter" -> "rotor diameters" (and "dependency"!) Changed
l16: "accurate" -> "accurately" Changed
l17: This sentence could be formulated more clearly. Maybe changing "to fill" to "in order to fill" or add a comma somewhere. Changed, we have used "in order to fill"
P8, line 25 : « affected » instead of « effected » Changed